# Automatic detection of avalanches using a combined array classification and localization

Matthias Heck[1], Alec van Herwijnen[1], Conny Hammer[2], Manuel Hobiger[2], Jürg Schweizer[1], and Donat Fäh[2]

[1]WSL Institute for Snow and Avalanche Research SLF, Davos
[2]Swiss Seismological Service SED, ETH Zurich, Zurich

**Correspondence:** Matthias Heck (matthias.heck@slf.ch)

**Abstract.**

We used a seismic monitoring system to automatically determine the avalanche activity at a remote field site near Davos, Switzerland. By using a recently developed approach based on hidden Markov models (HMMs), a machine learning algorithm, we were able to automatically identify avalanches in continuous seismic data by providing a single training event. Furthermore, we implemented an operational method to provide near real-time classification results. For the 2016-2017 winter period 117 events were then automatically identified. False classified events such as airplanes and local earthquakes were filtered using a new approach containing two additional classification steps. In a first step, we implemented a second HMM based classifier at a second array $14\,\mathrm{km}$ away to automatically identify airplanes and earthquakes. By cross-checking the results of both arrays we reduced the number of false classifications by about $50\,\%$. In a second step, we used multiple signal classifications (MUSIC), an array processing technique, to determine the direction of the source. Although avalanche events have a moving source character, only small changes of the source direction are common whereas false classifications showed large changes and thus were dismissed. From the 117 initially detected events during the 4-month period we were able to identify 90 false classifications based on these two additional steps. The avalanche activity based on the remaining 27 avalanche events was in line with visual observations performed in the region of Davos.

## 1 Introduction

During the winter seasons, snow avalanches are a common threat in mountain regions. Avalanche warning services therefore inform the public of the current avalanche danger. To assess the danger, warning services rely on information about the snowpack, amount of new snow, weather conditions and avalanche activity (McClung and Schaerer, 2006). Whereas the first three parameters can be measured or modeled, avalanche activity data are often hard to obtain, especially during snow storms or at night. Monitoring systems have therefore been developed to estimate the avalanche activity for a certain region.

Snow avalanches, like any other mass movement, generate seismic and infrasound waves (e.g. van Herwijnen and Schweizer, 2011b; Suriñach et al., 2005; Marchetti et al., 2015). Seismic signals of avalanches show some common characteristics, including a spindle shaped envelope of the time series (Nishimura and Izumi, 1997) and a typical frequency content between 2 and

30 Hz (Schaerer and Salway, 1980; Suriñach et al., 2001). Several classification approaches were therefore developed to automatically detect avalanches in seismic data. Leprettre et al. (1996) used a fuzzy logic approach to distinguish between different types of signals. Bessason et al. (2007) used a nearest neighbor approach to classify new recorded events. Using this approach, they were able to detect 65 % of all confirmed avalanches. Rubin et al. (2012) compared 12 machine learning algorithms, 10 of which were able to detect at least 90 % of all manually identified avalanches, however, at the cost of very high false alarm rates. Hammer et al. (2017) recently used hidden Markov models (HMMs), an advanced machine learning algorithm, to automatically detect large avalanches released during the winter of 1998-1999 in seismic data recorded by a single broadband station maintained by the Swiss Seismological Service (SED). Using this approach, they were able to identify 43 avalanches during a 5-day period within a radius of 30 km of the station. Heck et al. (2018a) also used the HMM approach to automatically detect avalanches, however, in data recorded during the winter season 2009-2010 by a seismic array consisting of seven less sensitive vertical geophones. They obtained the best results for the automatic detection by combining the classification results of all sensors and requiring a minimal event duration for the detections.

Apart from using seismic signals for the automatic detection of avalanches, several studies focused on the use of infrasound signals. Localization parameters determined using cross-correlation techniques were used to automatically identify avalanches in continuous data sets (Scott et al., 2007; Marchetti et al., 2015; Thüring et al., 2015). By comparing the back-azimuth with the directions of known avalanche paths, possible avalanche events were identified (Marchetti et al., 2015). Thüring et al. (2015) used a similar approach for the automatic detection, but relied on support vector machines (SVM), a machine learning algorithm.

In addition to the automatic detection of avalanches, Lacroix et al. (2012) and Heck et al. (2018b) used seismic array processing techniques to locate the source of the avalanche. Lacroix et al. (2012) implemented a beam-forming approach and were able to assign recorded avalanches to three known avalanche paths. Heck et al. (2018b) compared a beam-forming method with a multiple signal classification (MUSIC) approach (Schmidt, 1986) and obtained better results with the latter and they subsequently applied this method to avalanches monitored during a two-day period in March 2017. Based on these results they concluded that their seismic array mostly recorded infrasound due to the limited distance between the sensors. Nevertheless, they were able to reconstruct the avalanche path of several recorded events. Lacroix et al. (2012) and Heck et al. (2018b) both used less sensitive vertical component geophones for the seismic monitoring resulting in an avalanche detection distance of approximately 3 km.

Our aim is to automatically identify avalanches in continuous data recorded during the winter period 2016-2017 using the same machine learning techniques based on hidden Markov models as used by Heck et al. (2018a). To reduce the false alarm rate we first use an additional classification performed at a second array 14 km away to dismiss events recorded almost simultaneously at both arrays such as earthquakes and airplanes. In a second step, we analyze the median back-azimuth path of the detections using the MUSIC method as performed by Heck et al. (2018b) and dismiss all events with a randomly distributed back-azimuth. We performed the classification and localization of the events with the data recorded at the seismic array located in the Dischma Valley above Davos, Switzerland during the winter season 2016-2017 (yellow square in Figure 1). These results

were then combined with data obtained at the Wannengrat array, which is located $14\,\text{km}$ to the northwest of the Dischma field site (red square in Figure 1).

## 2 Field site and instrumentation

Prior to the 2016-2017 winter season, we installed two seismic arrays above Davos, Switzerland, similar to the systems described by van Herwijnen and Schweizer (2011a). The first array was deployed at the Dischma field site (yellow square in Figure 1), $14\,\text{km}$ away from Davos at the end of a tributary valley (Heck et al., 2018b). The field site is a flat meadow at an elevation of $2000\,\text{m}$ a.s.l. surrounded by mountain peaks which rise up to $3000\,\text{m}$. The second array was deployed at the Wannengrat field site above Davos at $2500\,\text{m}$ a.s.l. (red square in Figure 1). This field site is surrounded by several avalanche starting zones (van Herwijnen and Schweizer, 2011a).

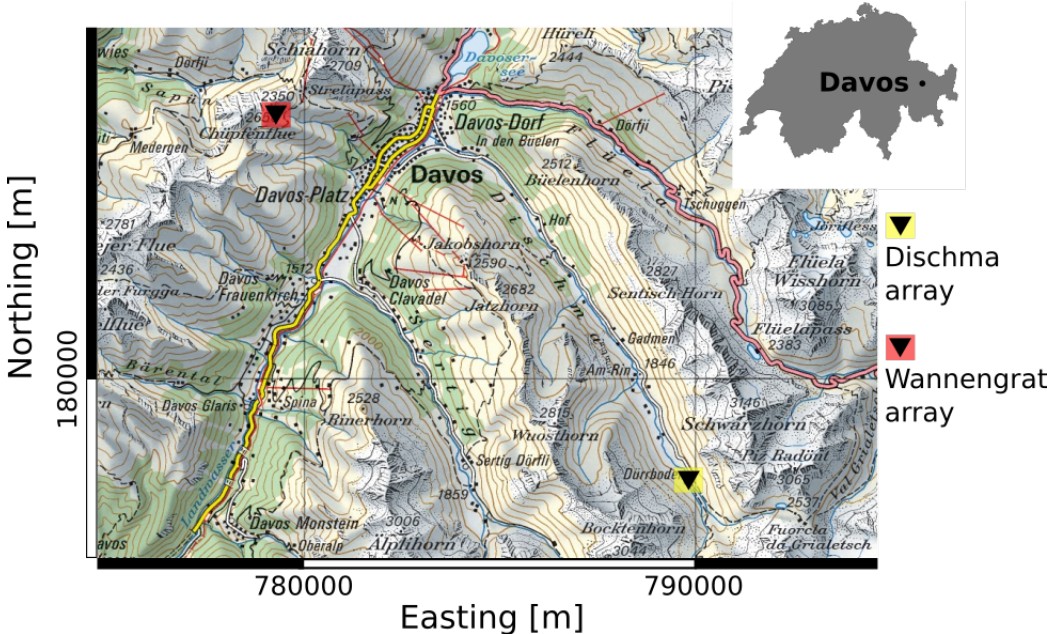

**Figure 1.** Map of the area of Davos, Switzerland. The two arrays are indicated by a black triangle on colored ground. Red represents the Wannengrat array, yellow the Dischma array. Reproduced by permission of swisstopo (JA100118).

Both arrays consisted of a $300\,\text{m}$ long string with 7 vertical component geophones with an eigenfrequency of $4.5\,\text{Hz}$. The sensors of the Dischma array were buried $50\,\text{cm}$ deep into the ground whereas the sensors at the Wannengrat field site were attached to rocks using an anchor. For each array the sensors were circularly arranged (Figure 2 a) and b). The maximum distance between two sensors at the Dischma and Wannengrat field site was $64\,\text{m}$ and $74\,\text{m}$, respectively, and the average distance was $36\,\text{m}$ at the Dischma array and $45\,\text{m}$ at the Wannengrat array.

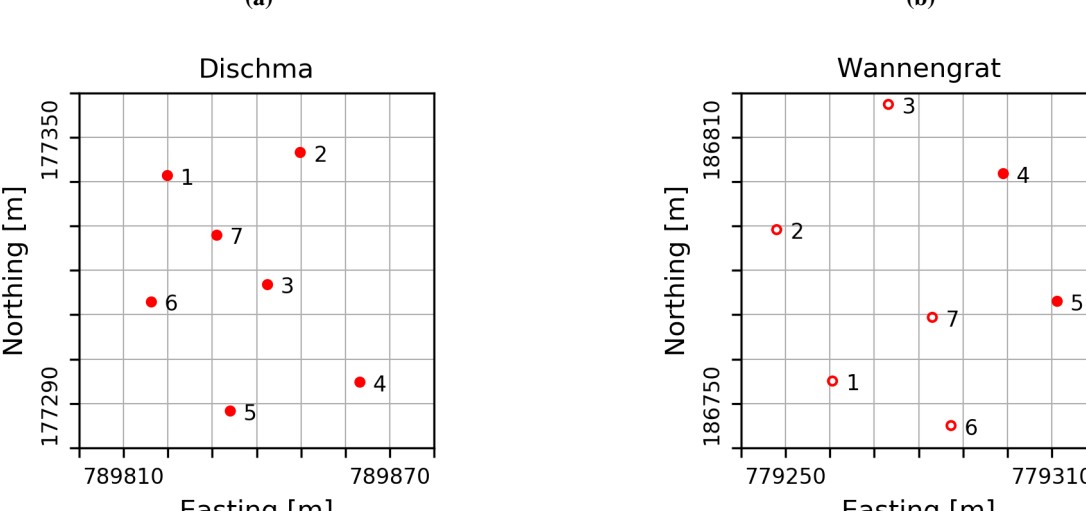

**Figure 2.** Setup of sensor arrays a) Dischma, b) Wannengrat. The open red circles indicate positions of not working sensors during the winter 2017.

The instrumentation and data logging systems were identical for both arrays. Data were continuously sampled at a rate of $500\,\text{Hz}$. However, due to technical problems only two sensors of the Wannengrat array recorded data throughout the entire winter (4 and 5 in Figure 2 b). Both field sites were equipped with several automatic weather stations (3 at Dischma, 4 at Wannengrat) as well as automatic cameras (8 at Dischma, 5 at Wannengrat). The automatic cameras visually monitored the surrounding slopes and images were recorded every 10 minutes throughout the winter (Figure 3). As already shown for avalanche activity periods in the winter season 2009-2010 by van Herwijnen and Schweizer (2011b), those images can help to identify and confirm seismic events produced by avalanches. In addition to the automatic cameras, we also performed field surveys shortly after a period of high avalanche activity to identify avalanches and map their outlines (Figure 4) (Heck et al., 2018b).

## 3 Methods

### 3.1 Data pre-processing

The continuous seismic data mostly consist of noise. Since for the current application noise is of little interest, we applied a simple threshold based event detector to reduce the amount of data (Heck et al., 2018a). For a window $i$ with a length of 1024 samples a mean absolute amplitude $A_i$ was determined. When $A_i \geq 5\overline{A}$, with $\overline{A}$ the daily mean amplitude, the data within the window were cut. If the amplitude threshold for the following window was also reached, data were concatenated. Furthermore, a section of $t = 60\,\text{s}$ was cut before and after the window to ensure that the onset and coda of each event was incorporated.

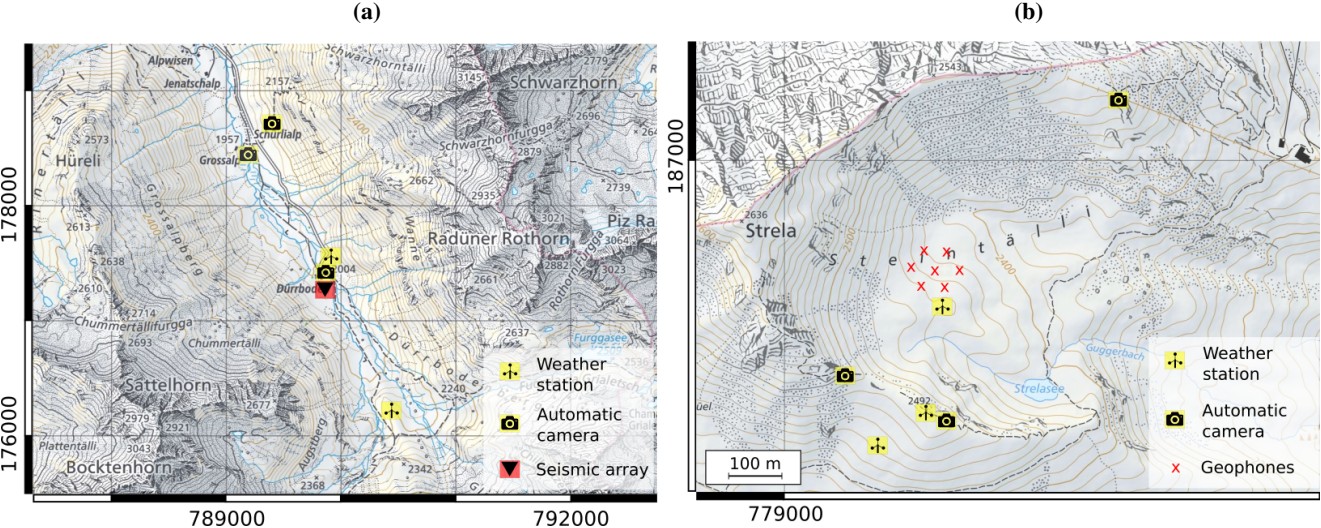

**Figure 3.** Detailed map of the field sites a) Dischma, b) Wannengrat. On the map the positions of the seismic sensors, automatic cameras and weather stations are indicated. Due to scaling not every weather station is shown on the map.

Doing so, data were reduced by $80\%$ to several data windows of various lengths. In addition, we filtered the data using a 4th order Butterworth bandpass filter with corner frequencies of 1 and $50\,\text{Hz}$.

## 3.2 Classification of events

To automatically identify avalanches in the continuous seismic data we used hidden Markov models (HMMs) (Rabiner, 1989). These statistical classifiers use a sequence of multivariate Gaussian probability distributions to model observations (e.g. seismic time series). To determine the characteristics of the distributions (i.e. mean and covariance) a large number of training sets of known events, so called pre-labeled training sets, are required. For each different type of observation (e.g. avalanche, airplane or earthquake in the seismic data) a separate HMM is trained. By combining all HMMs the whole classification system with several classes is constructed. This classical approach, which relies on a large number of well-known pre-labeled training sets, was successfully used to automatic identify seismic events in continuous seismic data (Ohrnberger, 2001; Beyreuther et al., 2012). Avalanches, however, are rare events and it is nearly impossible and too time consuming to obtain a large training set. To circumvent this, we performed the classification based on an approach developed by Hammer et al. (2012) exploiting the abundance of data containing mainly background signals to obtain general wave-field properties. From these properties a widespread background model can be learned. A new event model (e.g. representing avalanches) is then obtained by using the widespread background model to adjust the event model description by using only one training event. In contrast to the classical HMM approach, the classification system of this approach consists of a background model and one event model for each implemented event class. The classification process itself calculates the likelihood that an unknown data stream was generated by a specific event class for each individual HMM class (Hammer et al., 2012, 2013).

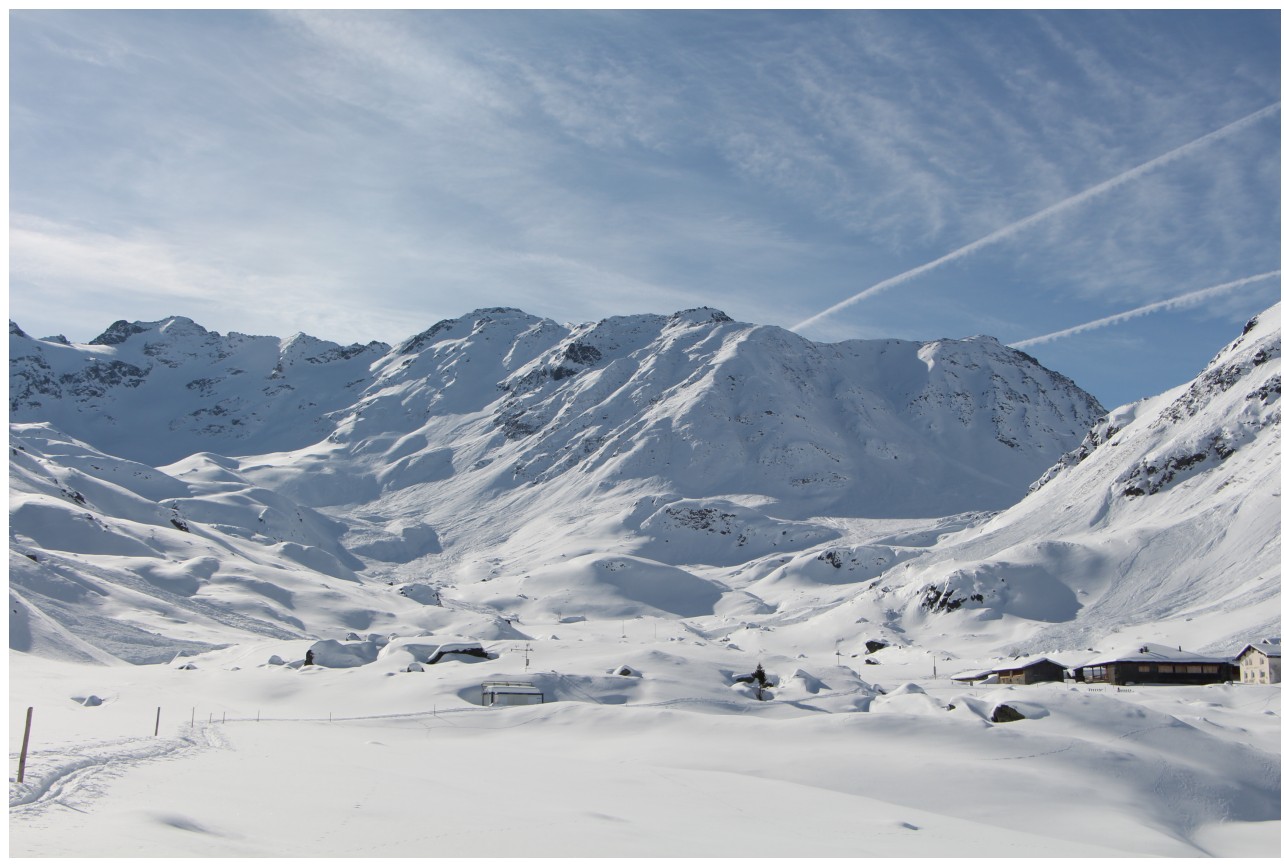

**Figure 4.** Picture of the field site of Dischma facing to the south. It was taken on 15 March 2017 shortly after a period of high avalanche activity and several recent avalanches were observed.

This new approach was successfully used on continuous seismic data collected at the Wannengrat field site during the 2009-2010 winter season by Heck et al. (2018a). Their classification, however, consisted of creating a new background model for each day and the resulting classification system was used to classify the data of the same day. A near real-time classification, as would be required for operational purposes, is then not possible. To overcome this problem, here we implemented a classification process by learning the background model using data from a different time window than the data we wanted to classify. To train the background model we used the pre-processed data taken from the window $t_{\mathrm{model}}$, whereas the pre-processed data we want to classify are in the time window $t_{\mathrm{class}}$ (Figure 5). This so-called operational classification was performed by using a window length of $t_{\mathrm{model}} = 24\,\mathrm{h}$ and $t_{\mathrm{class}} = 1\,\mathrm{h}$ with the start time of $t_{\mathrm{class}}$ corresponding the end of $t_{\mathrm{model}}$. This means, that our background model is always determined by the pre-processed data of a 24-hour window. By choosing a length of $1\,\mathrm{h}$ for the window $t_{\mathrm{class}}$, we were able to classify the pre-processed continuous seismic data of one hour during one step of the operational classification. Once one classification step, which is the classification of the window $t_{\mathrm{class}}$ is finished, both windows are shifted by one hour and the classification was executed for the shifted windows. The so performed classification takes $\sim 6\,\mathrm{min}$ for the

classification of one day without the feature calculation. In contrast, the classification performed by Heck et al. (2018a) only took $\sim 30\,\mathrm{s}$ for one day. All calculations were performed on a computer with a regularly available 8-core processor and $12\,\mathrm{GB}$ ram running a standard Ubuntu Linux Distribution.

As input for the HMMs a compressed form of the data was used, so-called features. Features represent different aspects of the time series such as spectral, temporal or polarization characteristics. Since we used single component geophones, we only used the following spectral and temporal features similar to those used by Heck et al. (2018a).

- – Central frequency (Barnes, 1993)

- – Dominant frequency (Kramer, 1996)

- – Instantaneous bandwidth (Barnes, 1993)

- – Instantaneous frequency (Taner et al., 1979)

- – Cepstral coefficients (Kanasewich, 1981)

- – Half-octave bands (Joswig, 1994)

For the feature calculation, we used a sliding window of width $w = 512$ samples and a step size of $0.05\,\mathrm{s}$ or 25 samples resulting in an overlap of $97\,\%$. We used in total 6 half-octave bands for the classification and the first half-octave band had a central frequency of $3.9\,\mathrm{Hz}$. Calculating the features from the pre-processed data takes $\sim 15\,\mathrm{min}$ for a complete day for all sensors. Since we shift the windows for $1\,\mathrm{h}$ after each step of the operational classification, the calculation of the features for a complete day needs to be performed only for the very first step. For the following steps it is sufficient to calculate only the features for the window $t_{\mathrm{class}}$ with a length of $1\,\mathrm{h}$, which approximately takes less than $\sim 2\,\mathrm{min}$ for the calculation.

The classification process consists of five steps: pre-processing, feature calculation, HMM construction, classification and post-processing (Figure 5). First the data used to build the background model are selected from the time window $t_{\mathrm{model}}$ and the data to be classified are determined by the window $t_{\mathrm{class}}$. The data from the selected time windows are pre-processed to reduce the amount of noise and then the features are calculated. In the HMM construction part, the features calculated from the data within $t_{\mathrm{model}}$ and from the data of the training event are used to construct a background model $\mathrm{HMM_{Back}}$ and an event model $\mathrm{HMM_{Event}}$. The event model $\mathrm{HMM_{Event}}$ was learned using only one training event that is representative of avalanches at a specific field site (Heck et al., 2018a). It was determined once and then applied for the entire winter season. In contrast, the background model $\mathrm{HMM_{Back}}$ was reconstructed every hour. Using both models, the pre-processed data within the window $t_{\mathrm{class}}$ were classified in the classification process. The features calculated for this window are therefore passed to the classifiers. Once the classification is performed, post-processing steps are applied as proposed by Heck et al. (2018a). The first step consisted of applying a duration threshold to the classified events. Each classified event shorter than $12\,\mathrm{s}$ in duration was dismissed. The second step, the so-called voting-based classification, combines the results of all sensors. Only events that were classified by at least 5 sensors are considered as possible avalanches.

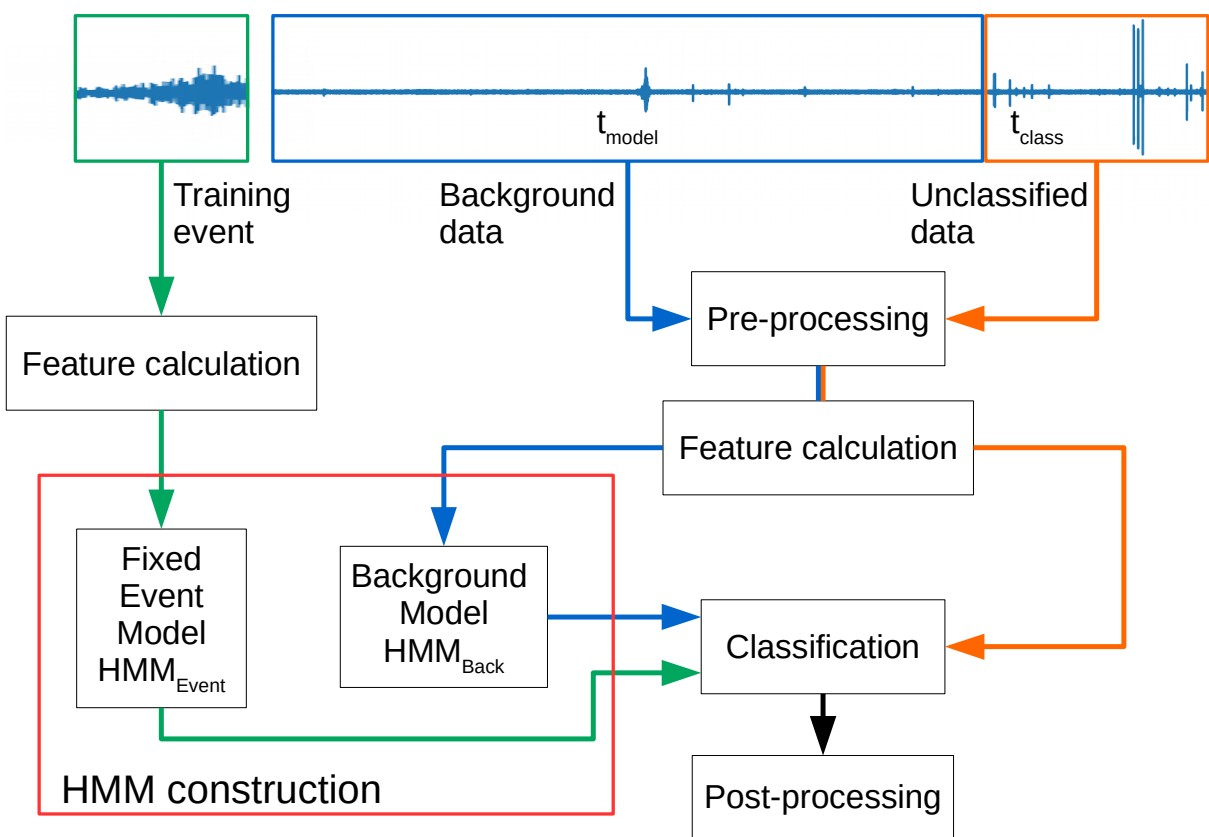

**Figure 5.** Flow chart of the classification process. Green lines show the construction of the event model. Blue lines show the construction of the background model. The orange lines show, how the data to be classified are processed.

## 3.3 Combined array detection

Initial classification results performed on the data set of the winter season 2016-2017 revealed that although the total number of detected events was low, many detected events were very likely generated by airplanes or regional earthquakes (local magnitude between 1.5 and 4 for earthquakes at local and regional distance triggered by at least 6 stations according to the earthquake catalog of the Swiss Seismological Service). In contrast to avalanches, which are recorded only at one array, these events are recorded at both arrays. We therefore used a combined array detection to remove earthquakes and airplanes from the detections. To perform the combined array detection, a second HMM was implemented to identify earthquakes and airplanes in the data recorded at the Wannengrat array at $14\,\mathrm{km}$ distance from the Dischma array. Classification results were then combined to remove all events recorded simultaneously at both arrays.

## 3.4 Localization results to confirm avalanches

Heck et al. (2018b) determined the direction of several avalanches using a multiple signal classification algorithm called MUSIC and were able to locate the avalanche path of several avalanche events based on the data of a single array. The MUSIC algorithm determines the back-azimuth angle and the apparent velocity of the incoming wave-field for a small time window. This time window is shifted to provide a time series of back-azimuth and apparent velocity values. The MUSIC method is based on the covariance matrix taking the data of all sensors into account at once, whereas beam-forming methods rely on pair-wise cross-correlation (Schmidt, 1986; Rost and Thomas, 2002). MUSIC can resolve multiple sources more easily than beam-forming methods. Furthermore, the MUSIC method can be applied to small frequency bands and the different frequency contents of the wave-field can be analyzed. For further information on multiple signal classification the reader is referred to Schmidt (1986) and Hobiger et al. (2016).

Heck et al. (2018b) found, that due to the small distance between the sensors, the seismic array mostly resolved sonic wave-fields rather than seismic wave-fields to estimate the back-azimuth. Using a median smoothing filter they then calculated a so-called median back-azimuth path with time. In this study, we used these median back-azimuth paths obtained by the event localization to decide whether a classification was associated with an avalanche. Specifically, we used a threshold value for the derivative of the median back-azimuth path. The assumption is that avalanche events have a smooth median back-azimuth path with little variations, whereas false detections have randomly distributed back-azimuths with large variations in time. By analyzing several avalanche events, especially the events identified by Heck et al. (2018b), we observed small changes below $10°$ for the back-azimuth path. Hence, we used a threshold value of $10°$ between two adjacent points of the median back-azimuth path for the event detection. Even for events passing close to the array, we observed changes below $10°$ in the median back-azimuth path. Heck et al. (2018a) suggested that a detected event should have a minimum duration of $12\,\mathrm{s}$ to be considered as an avalanche. For the localization step, however, it was necessary to increase this duration because the window length used for the median smoothing filter was already $8\,\mathrm{s}$ long (Heck et al., 2018b). To cover enough data points to use the minimal event duration as a reliable classification criterion, we therefore required a minimum length of $20\,\mathrm{s}$ for the back-azimuth path. Heck et al. (2018b) also showed that only the frequency content of the signal between 4.5 and $12.5\,\mathrm{Hz}$ contained information valuable for the localization performed at the used array. By further analysis of the already localized events by Heck et al. (2018b), we observed, that a reduced frequency range provided similar results. Hence we reduced the number of analyzed frequency bands to four bands between 6 and $7.5\,\mathrm{Hz}$ and were able to speed up the calculation time. Nevertheless, the processing time for the localization is about three times real time on the same computer used for the classification as mentioned earlier.

## 4 Classification results

We performed the avalanche detection for the data recorded at the Dischma array during the winter season 2016-2017 and compared the results with the avalanche activity visually obtained by local observers and compiled by the avalanche warning service at the SLF. It has to be noted that this compilation is incomplete and covers an area much larger than that monitored with the Dischma array. Therefore, comparison with this compilation remains indicative.

### 4.1 Overview of the winter season

The winter period of 2016-2017 was relatively short and characterized by a below-average snow depth. First snowfalls were quite late in the season, in the middle of December, followed by four weeks without substantial precipitation and low temperatures. Due to the constant high temperature gradient within the snowpack, a poorly bonded layer of depth hoar was formed at the base of the snow cover.

During the winter season, three significant snowfall periods occurred; one in each month from January to March (increase of blue line in Figure 6). Each of these snowfalls were associated with considerable avalanche activity in the region of Davos (red bars in Figure 6).

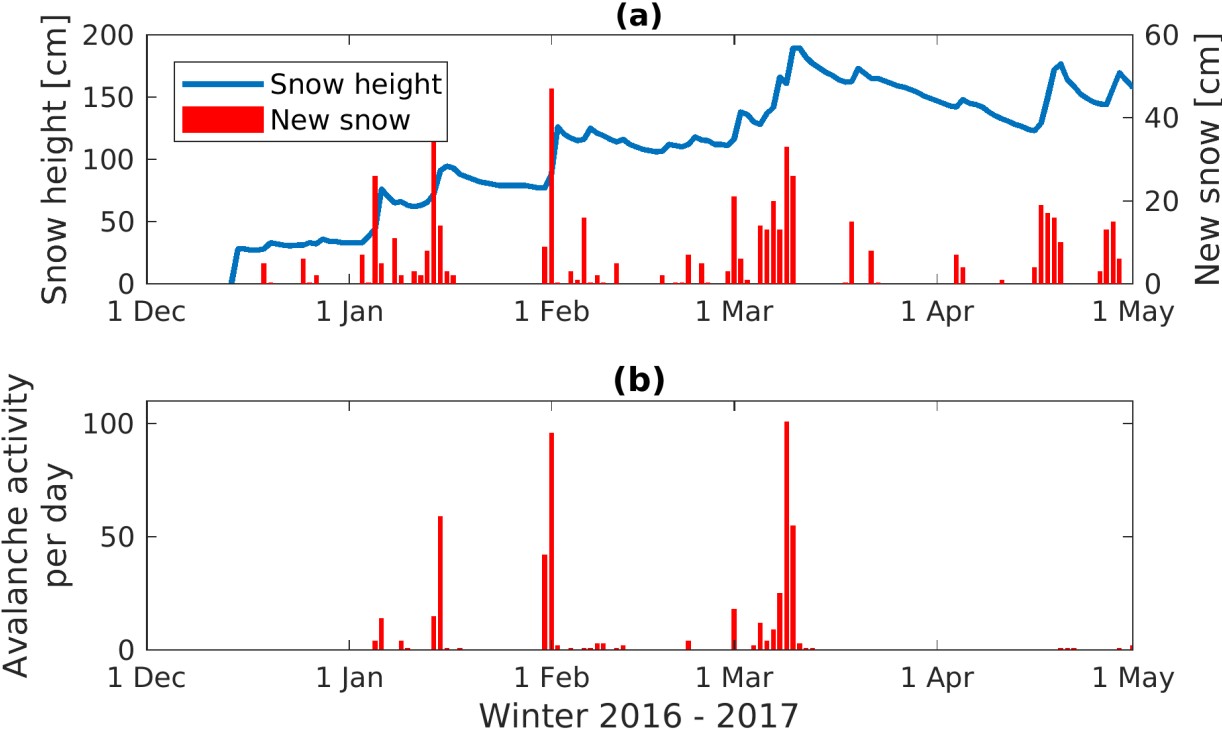

**Figure 6.** a) Snow height measured at the automatic weather station at Weissfluhjoch 12 km to the northwest of the Dischma array at $\sim 2600$ m a.s.l. for the winter season 2016-2017. Red bars are the height of new snow measured each day at 8:00 am. b) Number of avalanches observed per day in the region of Davos ($\sim 175$ km$^2$).

In addition to these avalanche observations, we analyzed the pictures taken by the automatic cameras installed at our field sites. Surprisingly, avalanche activity was low at the Dischma field site in January and February. During the early March snow storm the visibility was poor and only very few avalanches were identified on the images of the automatic camera. However,

once the storm was over, the intensity of the avalanche cycle became clear as many avalanche deposits were visible on the images. Five days after the storm we mapped 24 avalanches within a $4\,\mathrm{km}$ radius of the Dischma field site (Heck et al., 2018b).

## 4.2 Classification performed at single array

The main classification was performed at the Dischma field site for all seven sensors. Based on the visual inspection of the seismic data performed by Heck et al. (2018b), several avalanche events suitable as training events for the HMM were identified. However, they had mainly analyzed the period of high avalanche activity on 9 and 10 March 2017. Visually inspecting the entire winter season we identified 44 avalanche events. However, as already shown by Heck et al. (2018a), visually inspecting seismic data contains many uncertainties. An avalanche released on 9 March 2017 at 06:47 was already analyzed in detail by Heck et al. (2018b) and can unambiguously be classified as an avalanche. We therefore decided to use this event as our training event (Figure 7).

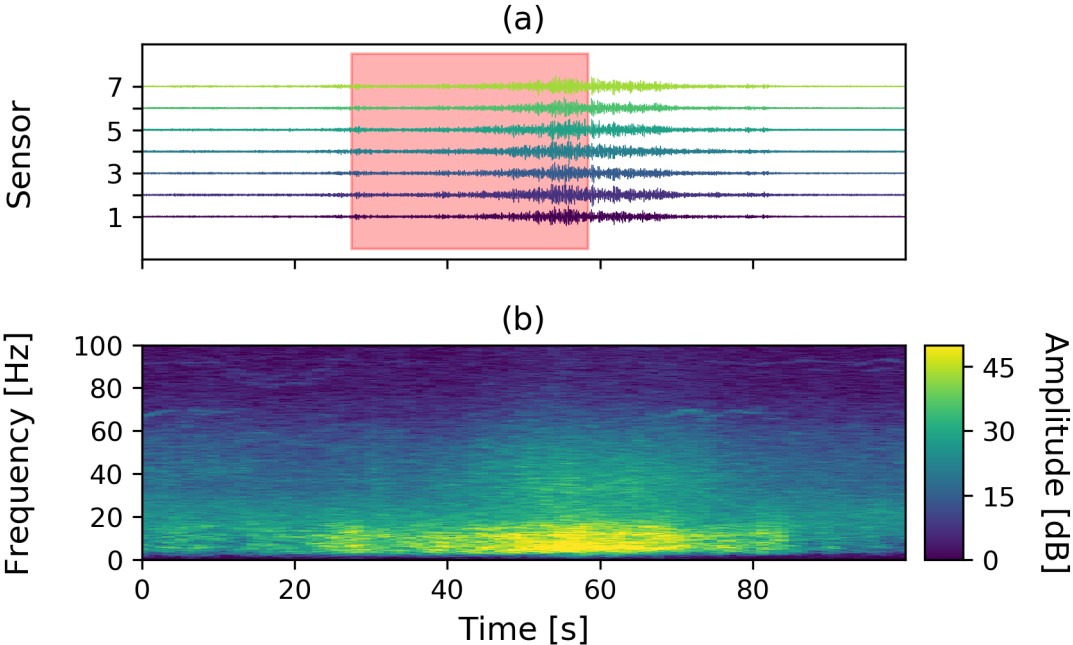

**Figure 7.** Avalanche released on 9 March 2017 at 06:47 used as training event for the classifier. a) time series for the 7 sensors. The red area indicates the part of the time series used as training event. b) corresponding spectrogram of the seismic time series.

Using the classifier trained with this event, we performed the classification for each single sensor of the array. In a next step, the results of the classification were post-processed; first all results of each sensor with a duration $\leq 12\,\mathrm{s}$ were dismissed (Heck et al., 2018a). Finally we dismissed all classifications that were classified by less than 5 sensors.

The classification and the following post-processing was applied to the continuous data set recorded from 1 January to 30 April 2017. For this period, a total of 117 events were classified as avalanches. A quarter of the events were detected by 5

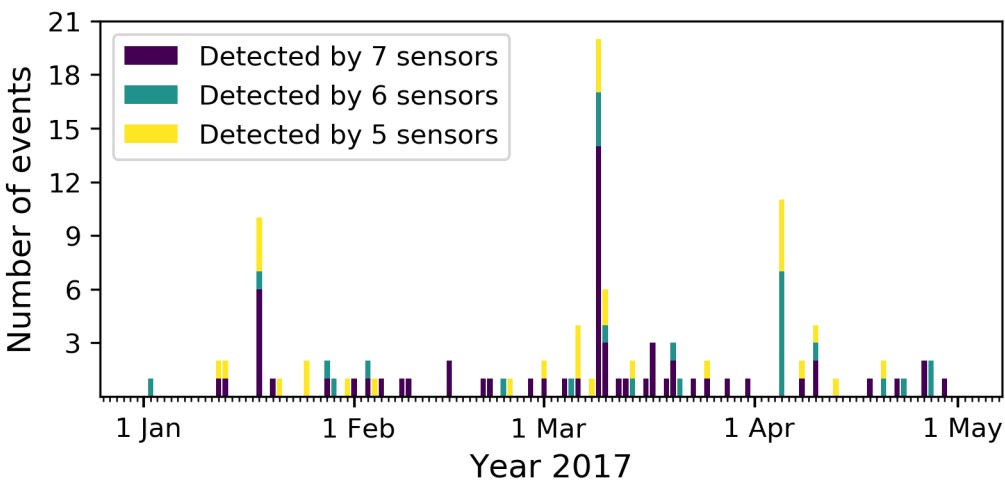

**Figure 8.** Classification results after post-processing (including voting-based classification) at the Dischma array. The colored bars indicate the number of classified events per day depending on the number of sensors: Violet bar indicates detections by 7 sensor, turquoise bars by 6 sensors and yellow bars by 5 sensors.

sensors, a quarter by 6 and about half by 7. Most events were classified during the early March snow storm on 9 and 10 March 2017 (Figure 8). In addition a peak in the middle of January and beginning of April and a cluster of events around the beginning of February was visible. The peak in January as well as the cluster around the beginning of February correspond with the avalanche activity period visually recorded in the region of Davos (Figure 6). For the peak in April, however, no avalanches
were observed in the surroundings of Davos. Furthermore, several single detections were distributed over the season showing no accordance with the visual avalanche observations. Therefore we visually inspected the time series and the corresponding spectrograms of each of the 117 classifications and found that the HMM also classified various airplanes (Figure 9 a)) and regional earthquakes (Figure 9 b) as avalanches.

Although these events can be distinguished from avalanches through visual inspection (e.g. the sharp onset visible for
earthquakes), the classifier identified these events as belonging to the avalanche class, even when we used different training events or varied the setup for the classification (results not shown). This was most likely because earthquake and airplane signals were more similar to avalanches than to the background model, and consequently tagged as avalanches.

Analyzing the features also showed the similarities of the different types of events, especially the time dependent behavior. In the beginning of the event, the feature behavior of the events is different, however, at the end of the event a similar
time dependent behavior is visible (Figure 10). Due to theses similarities, airplane and earthquake events are more similar to avalanches than to noise resulting in false classifications of these events.

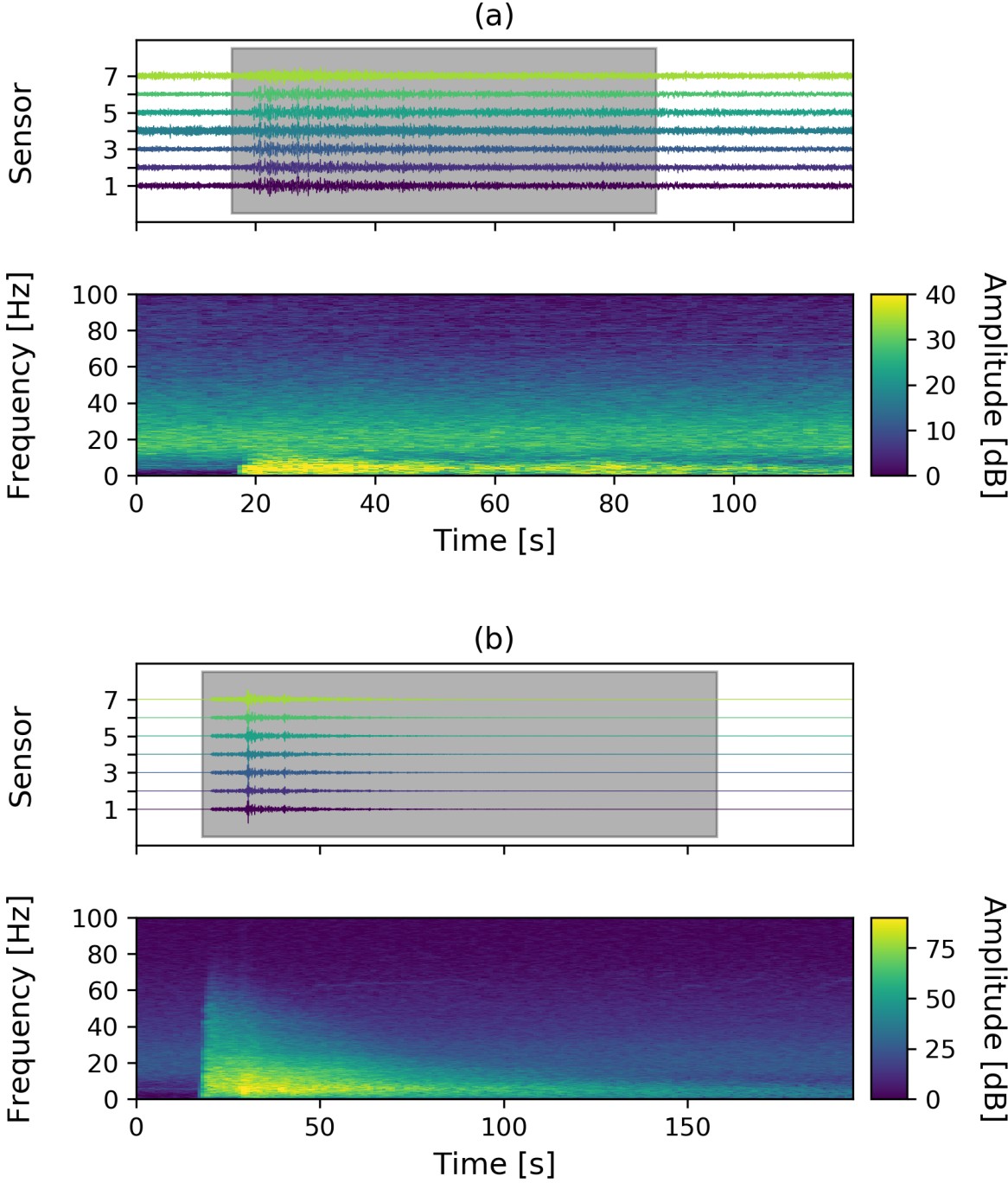

**Figure 9.** Time series and corresponding spectrograms for two false classifications, a) airplane, b) earthquake. The gray rectangular area indicates the part classified as event by the HMMs.

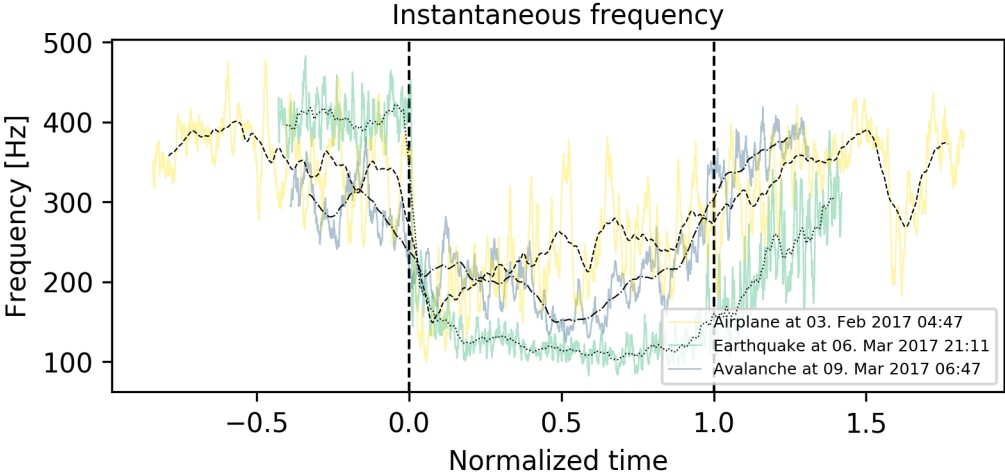

**Figure 10.** Feature instantaneous frequency for three different event types, yellow represents the signal produced by an airplane, green of an earthquake and blue of an avalanche. The black lines are the mean of the features. The dashed line at 0 indicates the start of the events and the dashed line at 1 the end of the event.

## 4.3 Classification performed at both arrays

The majority of the misclassifications were produced by two types of events, i.e. airplanes and earthquakes. A comparison of several detected earthquake events with the earthquake catalog of the Swiss Seismological Service (SED) showed, that all compared earthquake events occurred within a range of 120 km. As the Wannengrat array was deployed only 14 km away, all observed earthquakes were likely to be detected simultaneously at both arrays. Moreover, Davos lies within an approaching corridor of the international airport Zürich and several commercial airplanes pass by every hour at an altitude of at least 5 km. Similar to avalanches, airplanes also have a moving source character and due to the fast movement they are also observed almost simultaneously at both arrays. Indeed, a comparison of both time series revealed that earthquakes were recorded at both arrays at the same time whereas airplanes were recorded with a small delay of 20 to 30 s due to the movement of the source. The time series and spectrograms at Dischma and Wannengrat were very similar (Figure 11).

Avalanche signals, however, were only detected within a radius of 3 to 4 km of the array and were therefore only recorded at one array (Heck et al., 2018b). In order to eliminate classified events recorded at both arrays, we performed a second classification at the Wannengrat array. Due to similarities of the transient signals as mentioned earlier, a HMM trained with an airplane signal was capable to also detect earthquakes. A closer look at the classification results for the Wannengrat array revealed, that it was sufficient to only use the HMM trained with an airplane signal (not shown here). The number of detected events at this array varies strongly per day (blue bars in Figure 12).

The start times obtained by the secondary classification performed with the Wannengrat data were then compared with the classification results for the Dischma array. Overall, 53 of the 117 classifications were detected almost simultaneously at

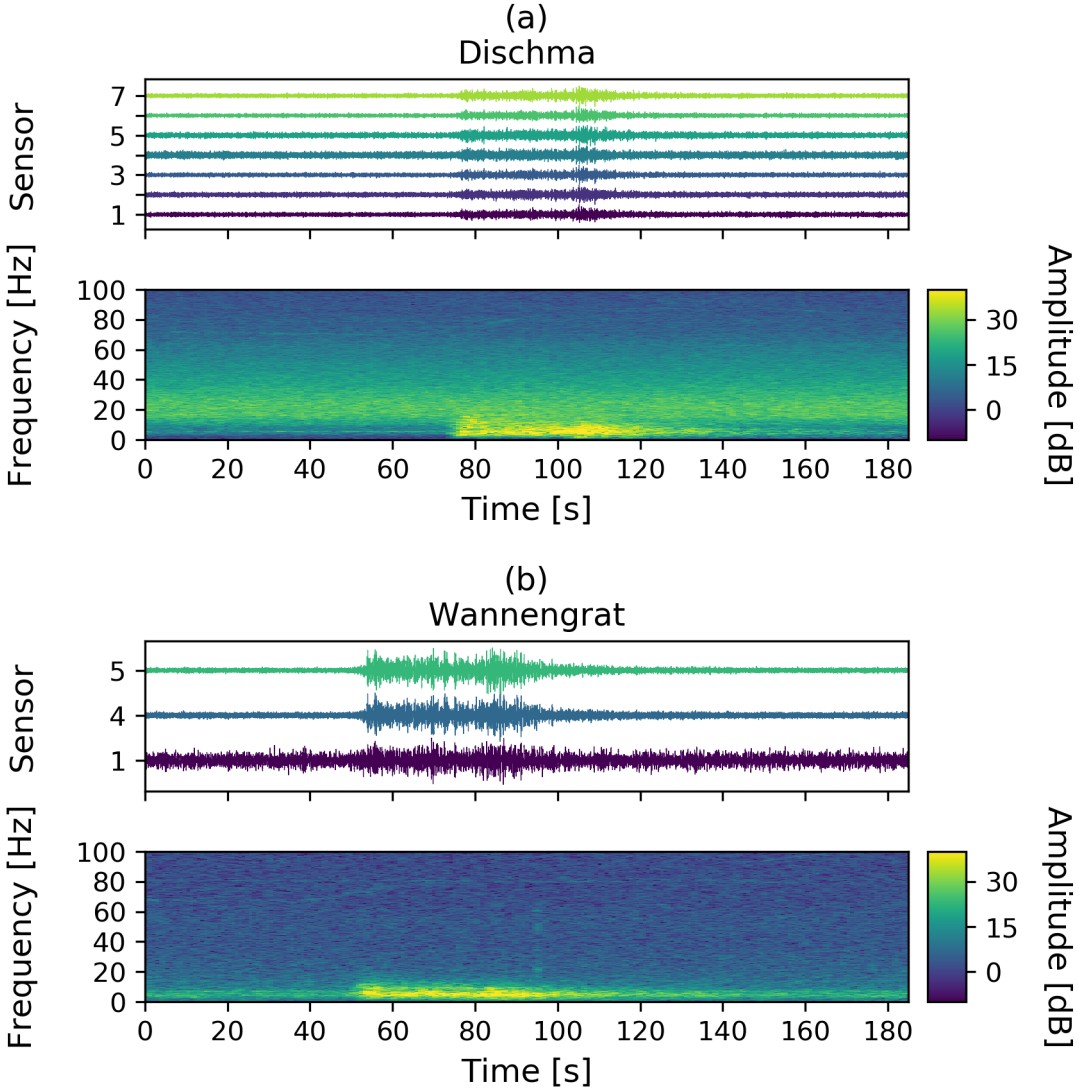

**Figure 11.** Time series and corresponding spectrograms of an airplane detected at both arrays on 28 January 2017 at 9:17. a) shows the signal recorded at the Dischma array and b) at the Wannengrat array.

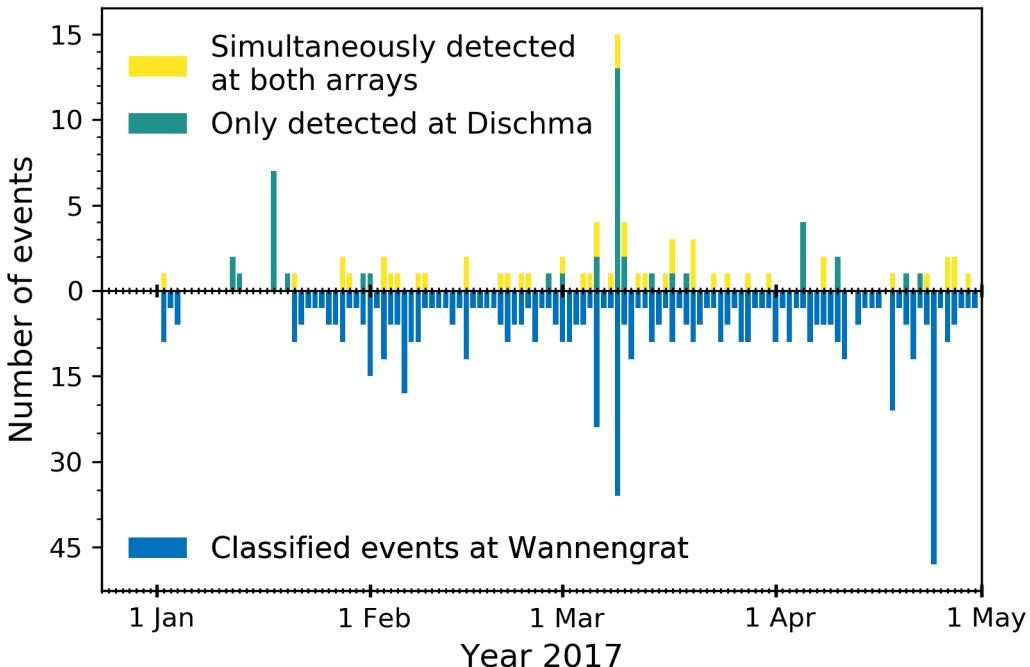

**Figure 12.** Yellow bars indicate the number of events detected at both arrays and turquoise bars only events recorded at the Dischma array. Blue bars are the number of airplanes and earthquakes detected at the Wannengrat array. Between 5 and 20 January no data were recorded at the Wannengrat array due to technical issues.

both arrays and we considered theses events as airplanes or earthquakes (yellow bars in Figure 12). After the comparison of the classification results obtained by both arrays 64 potential avalanche events remained (turquoise bars in Figure 12). The distribution of these events is similar to visually observed avalanches (Figure 6), except for detections at the beginning of April and no detections at the beginning of February. These events were only detected using the automatic classification approach.

5 Furthermore, due to the previously mentioned acquisition problem of the Wannengrat array, all events between 12 and 20 January 2017 were considered as avalanches as we had no further information from the second array. Hence, we expected some misclassifications among the remaining 64 avalanche events.

### 4.4 Localization post-processing

In a last processing step, we applied the MUSIC method to the remaining 64 classified events to estimate the back-azimuth
10 and to find a possible median back-azimuth path. The event used to train the HMMs had a duration of around $50\,s$ showing a median back-azimuth path with slight changes in the angle (straight black line in Figure 13 a). Before and after the event, however, the back-azimuths were randomly distributed as would be expected for noise. For the training event, the derivative

**(a)**

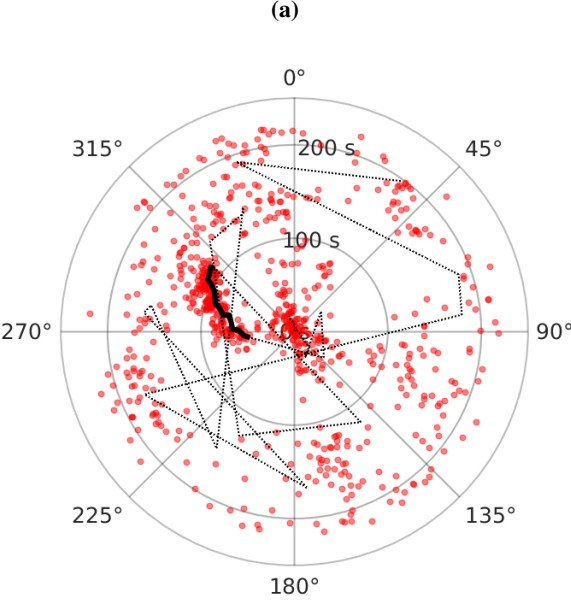

**(b)**

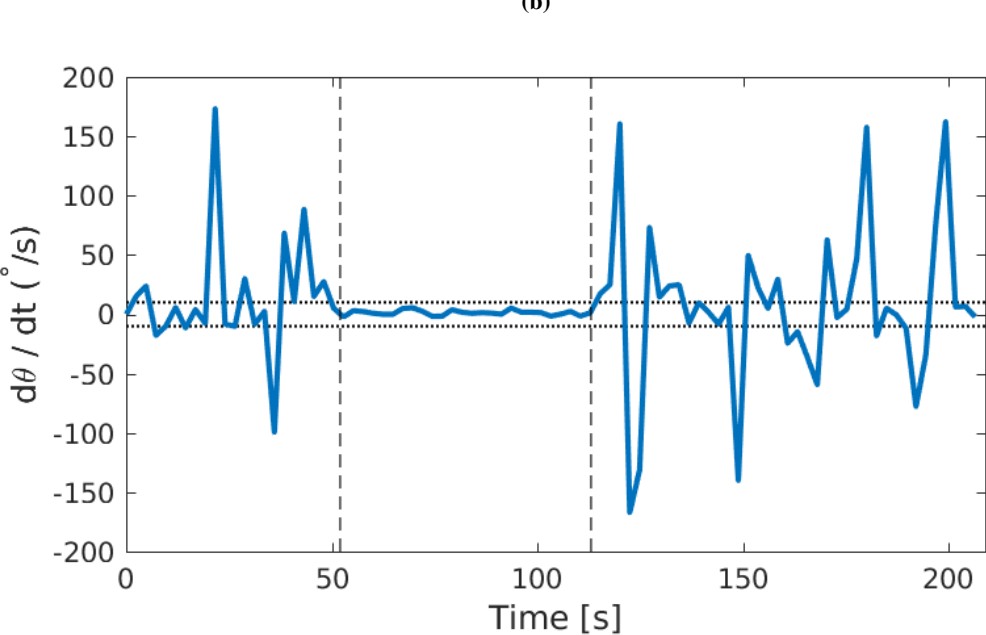

**Figure 13.** Localization results for an avalanche event recorded 9 March 2017 at 6:47. a) polar plot representation of the back-azimuth calculated using the MUSIC method. Red dots are the back-azimuth values for a single time window. The black line represents the median back-azimuth path. The solid part of the line has variations below the threshold value for the derivative, whereas the dotted line refers to strong variations. b) derivative of median back-azimuth path. The dotted lines represents the threshold value of $10°$. The part between $52\,\mathrm{s}$ and $113\,\mathrm{s}$ corresponds to the solid line in a).

of the back-azimuth path has low values for the $50\,\text{s}$ part with a median back-azimuth path with small changes (Figure 13 b). For this $50\,\text{s}$ long interval, changes below $10°$ were observed for the median back-azimuth path. Before and after the event, however, the changes are very high due to the randomly distributed back-azimuths. Further analyzed avalanche events also had changes of the back-azimuth below $10°$ and we therefore set the $10°$ as a maximum threshold value. Doing so, another 37 events

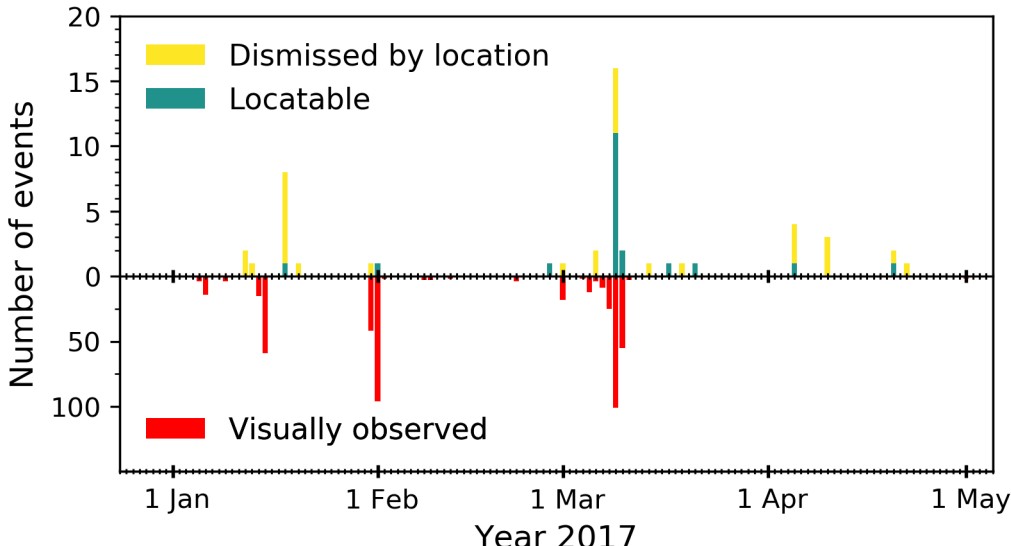

**Figure 14.** Turquoise bars are the number of events per day which are locatable and are considered as avalanches. Yellow bars are the number of events per day which were not locatable and therefore dismissed. Red bars are the number of avalanches visually observed in the area of Davos.

5 were dismissed and only 27 avalanche events remained. 15 of the remaining avalanche events were observed during 9 and 10 March 2017 and some events were detected during the other periods of considerable avalanche activity in February (Figure 6). Furthermore, another 10 single events were also confirmed. For each of the 27 events we determined a mean back-azimuth, which is the mean direction the signals were coming from. The mean back-azimuths were all pointing towards the surrounding slopes where we expected avalanches to release (Figure 15). Events with a duration longer than $100\,\text{s}$ were detected coming 10 either from the north-west or south-east.

Apart from analyzing only the events remaining after the combined array classification, we also performed the localization post-processing for those 53 events we had dismissed. Based on the localization, 48 events were again dismissed, but 5 had a median back-azimuth path within the threshold value. Hence, by directly applying the localization step 32 avalanche events remained, but including at least 5 false detections ($15\,\%$).

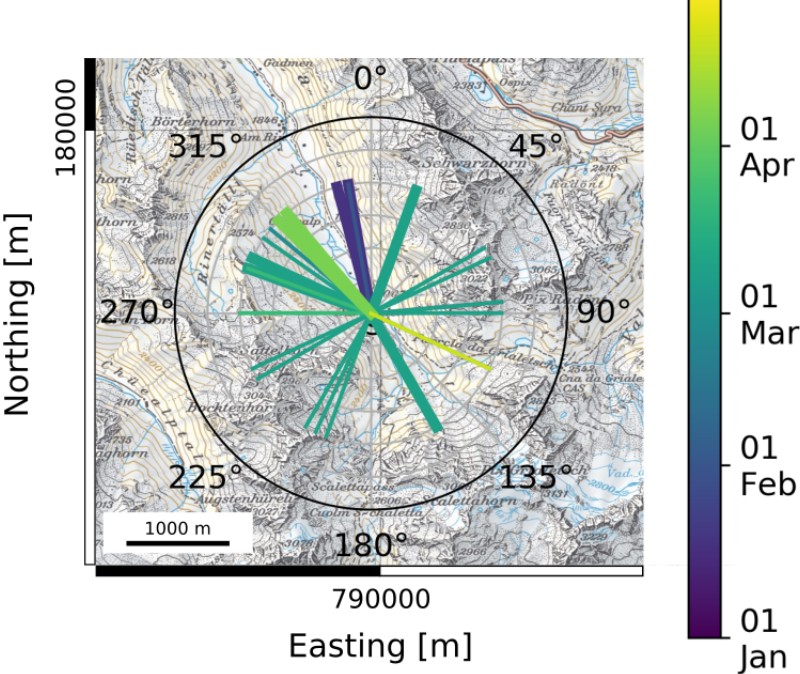

**Figure 15.** Polar plot representation overlaid on a map section of the field site. The angle represents the direction of the origin of the event, thin lines represent events with a duration $< 60\,\text{s}$, thick lines events with duration $\geq 60\,\text{s}$. The different colors of the lines represent the time of the year. Reproduced by permission of swisstopo (JA100118).

## 5 Discussion

We used hidden Markov models (HMMs), a machine learning algorithm, to automatically detect avalanches in data from seismic systems deployed above Davos, Switzerland. The approach builds on the work of Heck et al. (2018a), who adapted the HMM method developed by Hammer et al. (2017) to detect avalanches in continuous seismic data from a small aperture geophone array. Using their approach on our data resulted in automatic detections that still contained a large number of falsely classified events because only one event type (avalanche) and the background noise was used for the classification with HMM. Earthquake and airplane signals have characteristics closer to avalanches than the background noise, and were therefore included in the detections.

By combining the classification results with a classification performed at a second array located $14\,\text{km}$ away, simultaneously recorded events such as local earthquakes and airplanes could be dismissed. In addition, we applied the multiple signal classification (MUSIC) method to estimate the back-azimuth of the detected events to eliminate false alarms. Overall, this work flow allowed us to automatically identify 27 events that were very likely generated by avalanches, as the temporal trend

corresponded well with the avalanche activity for the region of Davos obtained through conventional visual field observations (Figure 6 and 14). It was not possible to confirm any event with visual observations since most avalanches released during periods of bad visibility. However, Heck et al. (2018b) manually identified 13 avalanches during 9 and 10 March 2017, 12 of which were automatically identified with the approach presented here.

Apart from HMMs, several other machine learning techniques are suited to classify signals in seismic data. It is possible to use a convolutional neural network for earthquake detection and location (Perol et al., 2018) or to pick the P-wave arrival of seismic wave fields (Ross et al., 2018). Comparable to the classical HMM approach, these studies rely on large pre-labelled training data sets. Another approach is the so-called Random Forest classifier, which can be used to discriminate seismic waves (Li et al., 2018). Automatic classification approaches are also suitable to differentiate between earthquakes and quarry blasts

(Hammer et al., 2013) or to characterize larger rockfalls (Dammeier et al., 2016). Further mass movements, such as landslides, can also be identified in the seismic data based on automatic classification approaches (Esposito et al., 2006; Hibert et al., 2014; Provost et al., 2016).

The automatic classification of avalanches yet remains a difficult task. Rubin et al. (2012) used several machine learning algorithms to identify avalanches in seismic data and compared the results obtained with the different approaches. With all

15 methods a high probability of detection was achieved, but the number of false alarms was too high. A recent study by Heck et al. (2018a) showed that HMMs are a suitable tool to detect avalanches, but there is still a need for additional post-processing steps. The work presented here confirms that HMMs in combination with further post-processing steps provide reliable classification results.

In addition, Heck et al. (2018a) highlighted the difficulty in obtaining a reliable classifier trained on data from a geophone

array very similar to the one used in this work. Their results showed that there were large differences in model performance between the sensors, with the number of detections per sensor ranging from about 150 to over 2000. This was attributed to local heterogeneities as the sensors were packed in a Styrofoam housing and inserted within the snowpack. Heck et al. (2018a) therefore suggested to deploy the sensors below the snow cover and either on or below the ground. In our deployment, the geophones were buried about half a meter below the ground on a flat meadow. This approach was successful as it resulted in a

much more consistent number of detections per sensor, ranging from 125 to 169. Clearly, the deployment strategy can have a substantial influence on the performance of the classifier.

In contrast to the classification approach used by Hammer et al. (2017), who used a fixed background model as they analyzed a relatively short period (5 days), we used an approach more suited for operational purposes. Indeed, for the operational set-up the background model was determined using 24 hours of data prior to the hourly data that were classified. In combination

with the post-processing steps related to signal duration and number of sensors the events were detected as suggested by Heck et al. (2018a), the HMMs identified 117 possible avalanche events (Figure 8). Even though this approach identified the main avalanche cycle in March 2017 (compare Figure 6 and Figure 8), visual inspection of the classified events indicated that at least $50\%$ of the events were false alarms produced by distant airplanes or regional earthquakes (Figure 9). Even by training a classifier with different feature combinations, changing the training event and/or the length of the training event,

the classification results did not substantially change and airplanes and earthquakes were still classified as avalanches. This

highlights the difficulty in training an accurate HMM for low energy signals generated by avalanches. We concluded that using HMMs to automatically identify avalanches in seismic data from our geophone array will inherently contain false detections, as the overall feature behavior from distant airplanes or regional earthquakes was very similar to signals generated by avalanches (Figure 10).

To circumvent the problem of developing an optimal event classifier for one specific array, we made use of a second array at the Wannengrat. There we performed a second classification to automatically identify airplanes and earthquakes using an event model trained by an airplane event. Since transient signals produced by earthquakes, airplanes or avalanches have similarities, the results obtained for the second classification based on the airplane event model also falsely identified avalanches and earthquakes. Hence, a classification performed with only one event model was sufficient. The assumption for the second

classification was that most falsely classified events were recorded at both arrays. Comparing the time series of detected events at both arrays allowed us to dismiss about $50\%$ of the classified events (Figure 12). Identifying co-detections across arrays is therefore an efficient approach to reduce the number of false alarms. Although it was possible, that the classification results of the second array contained avalanche events, it was unlikely that two avalanches released simultaneously at both field sites. Furthermore, avalanches were only recorded at one array since the distance between both arrays was about $14\,\mathrm{km}$. In the future,

a promising approach could be to reduce the distance to about 2 or 3 km, as this could also help improve the localization.

Although the combination of two arrays for the classification allowed us to reduce the number of false classification, some uncertainty remained about the origin of the identified events. In a final step, we therefore used the MUSIC method to estimate the median back-azimuth path, as suggested by Heck et al. (2018b), to further dismiss false detections. Similar approaches were suggested for the automatic detection of avalanches in infrasonic data by Marchetti et al. (2015) and Thüring et al. (2015). In

those studies, the back-azimuth of continuous infrasound data was calculated on the fly using cross-correlation techniques, and only events with slight changes in back-azimuth over a predefined minimal duration were assumed as avalanche events. In contrast, here we only determined the back-azimuth for events automatically identified by the HMM with the MUSIC method, as Heck et al. (2018b) showed that for our instrumentation pair-wise cross-correlation technique (beam forming) did not result in robust back-azimuth estimates. This last processing step further reduced the number of classified events to 27 (Figure 14).

We also found out, that the MUSIC method would have been sufficient to determine the reliability of a detection as it was not possible to locate airplane or earthquake events with our array. After applying the localization based step to all detections, 32 events were identified as avalanches, 5 more events compared to the combined array and the localization based classification.

After applying the combined array classification and the MUSIC method to the data, 27 classification remained for the winter season 2017. Nearly all of these classifications occurred during periods of observed high avalanche activity (Figure

14). However, for the first two periods of high avalanche activity in January and February only few events were detected, whereas in the surroundings of Davos many events were observed. This may be due to fact that the Dischma site is located about $12\,\mathrm{km}$ to the southeast of Davos and weather and snow conditions are sometimes different since major storms arrive from the northwest. Indeed, based on the images from the automatic cameras, very little avalanche activity was observed in the area in January and February. Nevertheless, it was possible to reconstruct the avalanche activity period in March based on the

automatic classification. Results from the localization showed that during the season avalanches released from many different

slopes at the field site (Figure 15). This could be observed especially during the snow storm in March. A seismic monitoring system is therefore a suitable tool to monitor a wide area and not just one single slope. Although the detection range is with 2 - 3 km rather small (Heck et al., 2018b) the seismic system in combination with an automatic classifier provides great potential to identify the major avalanche periods.

Although we were able to identify one major avalanche activity period in the winter season 2016 - 2017, the method presented here has its limitations. Based on the sensors used for the automatic monitoring, we identified avalanches within a range of 2 - 3 km. However, by using more sensitive sensors, e.g. seismological broadband stations, the detection range of avalanches can be increased, even up to 30 km for very large avalanches (run-out distance > 2 km (Hammer et al., 2017). However, it is difficult to deploy such sensors in mountain terrain, since these stations require existing infrastructure (e.g. electricity, storage room in a hut), which is typically not available at remote locations. In addition, the last post-processing step requires a second array. Hence low-power systems with less sensitivity proved to be the best solution. Furthermore, the limited power supply at the field sites also prevents performing first processing steps directly at the field sites and hence limits the possibility of near real-time analysis. However, it is possible to overcome this problem by designing special hardware for this particular task.

Based on the approach presented here, a near real-time classification of the seismic data and hence a near real-time detection of avalanches seems possible. The computational times on a computer with a regularly available 8-core processor with 12 GB ram are reasonably short and almost near real-time whereas the localization based on the MUSIC is very costly (three times real time). Although we decided to implement a combined array classification step to save computational time, directly localizing every detection is also possible. Since the amount of detections for the whole season was very low, a near real-time detection could be provided with or without the combined array classification. In future systems, pre-processing steps can be integrated in the data logging unit to reduce the amount of data while recording. Using a standard personal computer, feature calculation is performed near real-time for all sensors simultaneously as well as the HMM construction and the classification. However, a major obstacle of our method is the necessity of an adequate training event recorded at the seismic array. Using training events recorded at different arrays might be unreliable due to possible differences in the instrumentation and changes in the overall background noise or local heterogeneities in the local geology and in snow conditions. To set up the classification experts will still be needed to define correct and confirmed training events. Future research will assess the possibility to use one training event for several seasons recorded at the same array.

## 6  Conclusions

During the winter season 2016-2017 we used a seismic array to continuously monitor avalanche activity in a remote area above Davos, Switzerland. By implementing an operational classification method based on hidden Markov models (HMMs), we detected 117 events in the seismic data from January to April, which were likely produced by avalanches. Subsequent visual inspection revealed a false alarm rate of at least 50 %. Most of the false detections were associated with airplanes or earthquakes. By implementing additional steps such as a combined array classification and the localization of the events based on multiple signal classifications (MUSIC), we improved the classification results by reducing the number of identified events to 27. Only

using the localization to remove false detections resulted in at least $15\%$ of false detections yet at a higher computational cost. Our results therefore show that dismissing false detections with a second array improves the overall classification accuracy. If a second avalanche monitoring array is in the vicinity, combing the results of both arrays will improve the classification results. In future experiments we want to reduce the distance between the arrays to some kilometers to improve the localization of avalanches.

*Acknowledgements.* M.H. was supported by a grant of the Swiss National Science Foundation (200021_149329). We thank numerous colleagues from SLF for help with field work and maintaining the instrumentation.

*Competing interests.* The authors declare that they have no conflict of interest.

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
