# Peer review of "Automatic detection of avalanches using a combined array classification and localization"

_Earth Surface Dynamics, 2018_

## Referee Comment (RC1) · Anonymous Referee #1 · 1 Jun 2018

The authors present a method to automatically and in real-time classify avalanches in a region in Switzerland. They use an exemplary seismic waveform of an avalanche and use it to detect more avalanche events. However, this approach, just based on 1 array led to more than 50% of false detection of earthquakes and airplanes. They used a second array and removed signals that were simultaneously recorded at both arrays in order to reduce the number of false detection. However, even more false classifications were detect, when the authors computed back azimuths and removed the ones that scattered too much.

General comments: To me it seems that the approach they suggest is very difficult, despite they claim a near real-time detection of avalanches. So maybe it would be a better idea to e.g. choose windows with a high enough signal to noise ratio and then

perform the array processing in it rather than trying to find events with a master event, and the need for a second array and the array method? The authors discuss limitations of their method in the discussion. However, I feel that in this section the discussed literature is mainly their own papers i.e. first authors "Heck" and "Hammer". In addition I counted 11! references to the paper Heck et al. 2018B which is not published yet and it is therefore not possible to check the content, figures etc. of it. Could the authors provide the manuscript, as this manuscript seems crucial for the paper here?

Below are my detailed comment: p2, 5: what does "rather poor" mean? Can you quantify it or specify? Is this their conclusion or your interpretation? p2, 30: I think it is unclear what these arrays are. e.g. the one to locate avalanches and the one at 14 km distance and then your are talking about one in Dischma Valley and one at Wannengrat array. Are these the same arrays or different ones? Maybe the names or location of arrays should be introduced earlier and a link to figure 1 should be added? p2, 32: is this the winter season 2016/17? p3, 10: two ")" too much p3, 15: ")" too much p3, 15: where were these cameras and weather stations located? p4, 2: In this sentence you describe that the cameras helped to identify avalanches in the winter of 2016/17. But then you cite a publication from 2011? Clearly this publication does not describe the winter 2016/17? Maybe rephrase. p4, 6: Is the amplitude in noise that stable in time, that it ok to use a fixed threshold like you do or did you change it in time? p4, 6: given a sampling rate of 500 Hz your time window is only 2 seconds long when selected. This sounds pretty short to me when looking for avalanches. p5, 4 "Using these properties, a widespread background model can be learned from the general properties" I think this sentence sounds odd. Are you trying to built a model from information you derive from the general properties? p5, 4-7: I cannot follow how your method works in detail. Maybe the text should be rewritten with more reference to figure 3? p5, 14: are assuming that two events are separated by at least 24 hours? And if two events have a closer spacing in time they cannot be picked/ located? p5, 15: you state that the t_class window is 1 h long, but on p4, 6-8 you state that the chosen event is only 122 s long. Is there an error somewhere? p5, 25: What is a instantaneous

**ESurfD**
frequency? P5, 27: maybe you should explain cepstral coefficients? p5, 30: what do you mean with "the first half-octave band has [. . .] a total number of 6 bands"? p6, 6: what if the event model is very unlike the avalanche signal you try to detect? P6, 10: "Each classified event having a duration shorter than 12 s was dismissed" replace with "Each classified event shorter than 12 s in duration was dismissed" p7, 6: I am a bit surprised that you state that your second array at 14 km distance does not record the avalanche any more. After all you mentioned this array in the introduction that could detect avalanches up to a distance of 30 km (p2, 10). p7, 8: "12 km away" replace with "at 12 km distance" p7, 10: rephrase the heading as I find it pretty unspecific p7, 12: what MUSIC code did you use? Where is it available? p7, 27: does this approach not exclude avalanches along other potentially longer or more curved paths? p7, 33: what is the "used array"? p7, 34: "through further analysis" instead of "by further analysis"? p8, 2: "to speed up the calculation time": you "reduce the calculation time" or "speed up the calculation" p8, 2: so if I understand this correctly for a 2 minute long window it takes 6 minutes to process? So in order to do this in real time you need to skip time windows e.g. of "noise" p8, 15: figure 4a p8, 16: figure 4b p9, figure 4: maybe remove the legend in figure 4b as the information is already there as label of the y axis. Could you limit the yaxis at 110 or so in order to make the low numbers of avalanches in February more visible? p9, 2: On p7, 30 you state that you minimum event length is 20s whereas here you state it is 12 s. p9, 6: What do you mean with "classes with 5 and 6 votes" what votes? p9, 10: It that a good thing or a bad thing that you detect avalanches that are not listed in figure 4? E.g. does this mean that there are avalanches missing in figure 4 that should have been listed or are there completely different avalanches recorded in different areas and the only common thing is the huge amount of snow in that time period? p10, figure 5: move the sentence "the red area..." up to the description of subfigure a p10, figure 6: what do you mean with vote in the legend? What is a vote in the context of avalanches? p11, 2: two ")" too much p11, 21: I keep wondering why you detect the avalanched only up to 4 km distance and not 30 km distance as mentioned in the introduction. p11, 31: "except for detections at the

beginning of April" consider replacing it with: "except for detections at the beginning of April and no detections at the beginning of February" Any ideas as of why these avalanches in February could not be detected? p12, figure 7: Does the gray area in figure 7 have the same meaning as the red area in figure 5? If yes I suggest to use the same color. p12, figure 7: I am surprised about the low frequency content of the airplane and the lack of overtones. How did you classify this as an airplane? p13, figure 8: What is the unit of the normalised time and how is it calculated? Do the events have the same length or did you just stretch/ squeeze them to fit in between 0 and 1? p13, 6: one ")" too much p13, 8: one ")" too much p13, 11: Do you know what these 37 other avalanche like events might be? Maybe these are just avalanches along an unexpected path or longer paths? p13, 16: It sounds to me a bit like you remove events until you end up with back azimuths or locations you would like to get. p14, figure 9: How do you know that these are airplanes? p15, discussion: I find the discussion a bit repetitive with respect to the rest of the manuscript. Many points seem to have been made already in the rest of the text. Also my impression is that they barely refer to work of others in the discussion i.e. papers that are not lead by "Heck" or "Hammer". p15, figure 10: change to that the legend is not overlapping the bar any more p16, figure 11: "for avalanche event" replace with "for an avalanche event" Figure 11a: I don't understand to what part of the figure you refer to with "solid part". Beneath what threshold? Figure 11b: is this really the derivative of the angle (y axis label) or derivative of the back-azimuth path (caption)? To me this figure seems to show the "angle" or "back azimuth" during, before and after the avalanche event with very stable back azimuths during the event and larger scatter afterwards. p17, figure 12: so there are 100 visually observed avalanches in Davos but you could detect only 20? Were you too far away or was this recorded but not classified as event? Move the legend so that it does not overlap with the bars p17, 1: "closer" replace with "closer to"? p17, 8-10: First you say that you could confirm no avalanche visually, but in the next sentence you state that "another 12" events were identified. Were they identified in a different way i.e. not visually or is there an error in the sentence? P18, figure 13:

How do you know to what distance the duration of the event corresponds to? p18, 4: number of votes: in my opinion it would be better to replace "vote" with something like "detections on sensors" or similar. p18, 12: the overall feature behavior from distance airplanes... "was" not "were" p19, 9: remove "really". Based on the 5 events that were possible to locate, it is apparently possible to detect some avalanches on both arrays. p19, 9: I am not sure I fully agree. It is not possible to record an avalanche at 14 km distance if it couples to the ground sufficiently or is large enough? p19, 10: "since distance" replace with "since the distance" p19, 10: I am not sure where installing two arrays at 2-3 km distance would help. They would then pick up the same avalanches, and hence "events recorded at both arrays" are then not a valid criteria any more to find falsely classified earthquakes or airplanes... p19, 11: "improving" replace with "improve" p19, 22-24: Can you not locate airplanes and earthquakes with the array because the frequency content is different? So if the MUSIC method is perfectly suitable of detecting avalanches, why should one go through the hassle of finding a exemplary event, the need of having two arrays and then removing a lot of false detections? Rather than using the output from the array method to detect evetns? p19, 26: typo in "theses" p19, 30: typo "form" p19, 32: "avalanches were released" instead of "avalanches released"? p20, 5: Why is it that costly? Can the processing be sped up? p20, 14: "be still needed" replace with "still be needed" p21, references. There are 11! referrals in the text to a not published paper (Heck et al. 2018b). Can the authors provide the manuscript in order to cross-check e.g. the content?

---

## Referee Comment (RC2) · Anonymous Referee #2 · 22 Jun 2018

The paper by Heck et al. proposes a seismic methods to automatically determine the avalanche activity at a remote field site. Avalanches are automatically identified using a machine learning algorithm based on Hidden Markov Models (HMMs) applied to a little training dataset. The number of false detections was significantly reduced through two additional classification steps: (i) an additional HMM based classifier at a second array located 14km away to identify airplanes and earthquakes; (ii) the identification of the direction of the source. From the 117 initially detected events during a 4 month period were identified 90 false classifications based on these two additional steps. The obtained avalanche activity based on the remaining 27 avalanche events was in line with visual observations. The paper is well shaped and proposes a promising step forward in the field of seismic characterization of snow avalanches.

[Figure]

I'd like the author to better address the possible technical limitations of their methods, in particular the field deployment and the near-real-time application of the classification methods based on two stages. The HMM application to the seismic network object of the study is used to identify events and to filter possible false detection using a directional criterion. The authors already state that the computational time is reasonably short and almost near real-time whereas the localization would be very costly (three times real time). Maybe there is not a chance to perform a simplified, possibly faster directional classification based on few sensors and not on the whole seismic array. In addition, I expect that the network geometry has a strong impact on the success rate of this latter criterion, could you add some details on that? Then a second, distant seismic array is used to filter simultaneous signals produced by anthropic sources or earthquake. I have the impression that this second stage can be surely useful to recognize earthquakes but it probably needs a calibration for anthropic sources. In addition, technical limitations in such extreme environments like high Alpine areas (e.g., data transmission) can be a possible trivial but concrete limitation for a real time application.

The application of the proposed methodology on another dataset gathered on another test site would be of great interest for the reader. For instance, is it possible to run the methods the other way round, testing it on the other array currently used for the second classification step?

Visual observations are used as validation, could the authors add some information about that? Which are the observation sources? How is compiled the avalanche catalog? If available, an image of one reference event could be also useful to show the test site.

Figure 2, it would be useful to add a map with terrain information (slope, morphology, etc).

---

## Author Comment (AC1) · 17 Oct 2018

**Reply to reviewer 1**

We thank the reviewer for the constructive comments. Below we reply in detail.

Bessason et al. obtained a propability of detection around 65%. Seismic signals generated by rockfalls and debris flows were wrongly classified as avalanches and vice versa. Information about the false alarm rate was not given directly, but seems to have been high. (P.2 L.3)

*Bessason et al. 2007 used a nearest neighbor approach to classify newly recorded events. Using this approach, they were able to detect 65% of all confirmed avalanches.*

Thanks for pointing this out. We rephrased this sentence and added a reference to Figure 1. (P.2 L.34)

*We performed the classification and localization of the events with the data recorded at the seismic array located in the Dischma Valley above Davos, Switzerland during the winter season 2017 (yellow square in Figure 1). These results were then combined with data obtained at the Wannengrat array, which is located 14 km to the northwest of the Dischma field site (red square in Figure 1).*

Yes, we referred to the winter season 2016-2017. We changed the number in the text.

Removed.

Removed.

We included a detailed view of both field sites including all different types of weather stations and automatic cameras.

We rephrased this sentence to clarify that van Herwijnen et al. performed a similar analysis in 2011 in the surroundings of Davos. (P.4 L.5)

*As already shown for avalanche activity periods in the winter season 2009-2010 by van Herwijnen et al. 2011b, those images can help to identify and confirm seismic events produced by avalanches.*

P4, 6: Is the amplitude in noise that stable in time, that it ok to use a fixed threshold like you do or did you change it in time?
As mentioned in the text, we use a threshold value that is 5 times higher than the daily mean amplitude, i.e. the noise amplitude threshold changes each day.

P4, 6: given a sampling rate of 500 Hz your time window is only 2 seconds long when selected. This sounds pretty short to me when looking for avalanches.
In this step we are looking at the seismic energy recorded at the sensors for a 2 second window. If the energy for this window is higher than the threshold, we look at the following windows. If the energy for these windows also reaches the threshold value, we concatenate the windows. As soon as the energy decreased below the threshold value, we cut an additional 60 s before and after the total length of the event to not dismiss the onset and the coda of the event.
We clarified this in the text. (P.4 L.13)

*For a window i with a length of 1024 samples a mean absolute amplitude $A_i$ was determined. When $A_i <= 5 A^*$, with $A^*$ the daily mean amplitude, the data within the window were cut. If the amplitude threshold for the following window was also reached, data were concatenated. Furthermore, a section of t=60 s was cut before and after the window to ensure that the onset and coda of each event was incorporated. Doing so, data were reduced by 80% to several data windows of various lengths.*

P5, 4 "Using these properties, a widespread background model can be learned from the general properties" I think this sentence sounds odd. Are you trying to built a model from information you derive from the general properties?
We rephrased the sentence. (P. 5 L.14)

*From these properties a widespread background model can be learned.*

P5, 4-7: I cannot follow how your method works in detail. Maybe the text should be rewritten with more reference to figure 3?
We rephrased the section of the brief introduction of the Hidden Markov Models. However, this topic is rather complex and we provided references. (P.5 L.4)

*To automatically identify avalanches in the continuous seismic data we used hidden Markov models (HMMs). These statistical classifiers use a sequence of multivariate Gaussian probability distributions to model observations (e.g. seismic time series). To determine the characteristics of the distributions (i.e. mean and covariance) a large number of training sets of known events, so called pre-labeled training sets, are required. For each different type of observation (e.g. avalanche, airplane or earthquake in the seismic data) a separate HMM is trained. By combining all HMMs the whole classification system with several classes is constructed. This classical approach, which relies on a large number of well-known pre-labeled training sets, was successfully used to automatic identify seismic events in continuous seismic data (Orhnberger 2001, Beyreuther et al. 2012).*

*Avalanches, however, are rare events and it is nearly impossible and too time consuming to obtain a large training set.*
*To circumvent this, we performed the classification based on an approach developed by Hammer et al. (2012) exploiting the abundance of data containing mainly background signals to obtain general wave-field properties. From these properties a widespread background model can be learned. A new event model (e.g. representing avalanches) is obtained by using the background model to adjust the event model description by using only one training event. In contrast to the classical HMM approach, the classification system of this approach consists of a background model and one event model for each implemented event [or observation]? class.*
*The classification process itself calculates the likelihood that an unknown data stream was generated by a specific event [or observation]? class for each individual HMM class (Hammer et al. 2012, Hammer et al. 2013).*

P5, 14: are assuming that two events are separated by at least 24 hours? And if two events have a closer spacing in time they cannot be picked/ located?
We do not assume that two events are separated by at least 24 hours. We just use the data within a 24-hour window to construct our background model and a one hour window is then classified. Once the classification is performed, we shift both windows one hour.
We added additional information to clarify this. (P.6 L.8)

*This means, that our background model is always determined by the pre-processed data of a 24-hour window. By choosing a length of 1 h for the window $t_{class}$, we were able to classify the pre-processed continuous seismic data of one hour during one step of the operational classification. Once one classification step, which is the classification of the window $t_{class}$ is finished, both windows are shifted by one hour and the classification was executed for the shifted windows.*

P5, 15: you state that the t_class window is 1 h long, but on p4, 6-8 you state that the chosen event is only 122 s long. Is there an error somewhere?
t_class is the length of the window we want to classify, which means that we want to identify all events within this window and define their origin using the HMM. The length of the identified events is independent of t_class.

P5, 25: What is an instantaneous frequency?
The instantaneous frequency is the time dependent change of phase.
Taner, M., F. Koehler, and R. Sheriff (1979). Complex seismic trace analysis, Geophysics 44, no. 6, 1041–1063. We now include this reference in the text.

P5, 27: maybe you should explain cepstral coefficients?
We added a reference for the cepstral coefficients. (P.7 L.7)

P5, 30: what do you mean with "the first half-octave band has [. . .] a total number of 6 bands"?
In total, we calculated 6 half-octave bands. The first half-octave band had a central frequency of 3.9 Hz. Depending on this frequency, the central frequency for the remaining half-octave bands are already determined. We rephrased this sentence. (P.7 L.14)

*We used in total 6 half-octave bands for the classification and the first half-octave band had a central frequency of 3.9 Hz.*

P6, 6: what if the event model is very unlike the avalanche signal you try to detect?
Then we would not be able to identify avalanches in the winter season. Therefore, it is crucial to use a training event which is representative of avalanches at this specific field site. We changed the wording in the text to emphasize the importance of the event model and included a reference to Heck et al. (2018). (P.7 L.24)

*The event model HMM$_{Event}$ was learned using only one training event that is representative of avalanches at a specific field site (Heck et al. 2018a). It was determined once and then applied for the entire winter season.*

P6, 10: "Each classified event having a duration shorter than 12 s was dismissed" replace with "Each classified event shorter than 12 s in duration was dismissed"
We rephrased this sentence.

p7, 6: I am a bit surprised that you state that your second array at 14 km distance does not record the avalanche any more. After all you mentioned this array in the introduction that could detect avalanches up to a distance of 30 km (p2, 10).
The detection radius strongly depends on the size of the avalanche, and to a lesser degree on the sensors used to record the signals. It is possible to identify avalanche within a range of 30 km, however, only by using broadband seismometer and only for very large events (runout distances > 2000 m). These events are very uncommon and were not observed during our study period. The avalanches we monitored were substantially smaller avalanches and we used less sensitive geophones. For these avalanches, we recently showed that the detection radius is about 3 km (Heck et al., 2018b).

p7, 8: "12 km away" replace with "at 12 km distance"
We rephrased this sentence.

p7, 10: rephrase the heading as I find it pretty unspecific
We rephrased the heading. (P.9 L.1)

*Localization results to confirm avalanches*

p7, 12: what MUSIC code did you use? Where is it available?
The MUSIC code we used was developed by Dr. Manuel Hobiger from the Swiss Seismological Service. We will supply the scripts and data on an open access storage.

p7, 27: does this approach not exclude avalanches along other potentially longer or more curved paths?
We analyzed one event which approached the array to less than 100 m and therefore exhibited large changes in the back-azimuth. These changes were still well within the limits we used here. For more distant avalanches, changes in back-azimuth will be much smaller, even for more curved paths.

p7, 33: what is the "used array"?
The used array is also the Dischma array. We rephrased this sentence. (P.11 L.4)
*… performed at the Dischma array.*

p7, 34: "through further analysis" instead of "by further analysis"?
Changed as suggested.

p8, 2: "to speed up the calculation time": you "reduce the calculation time" or "speed up the calculation"
We rephrased the sentence.

p8, 2: so if I understand this correctly for a 2 minute long window it takes 6 minutes to process? So in order to do this in real time you need to skip time windows e.g. of "noise"
That is correct. This is also one reason to pre-process the data.

p8, 15: figure 4a p8, 16: figure 4b p9, figure 4: maybe remove the legend in figure 4b as the information is already there as label of the y axis. Could you limit the yaxis at 110 or so in order to make the low numbers of avalanches in February more visible?
We changed the figures as suggested.

p9, 2: On p7, 30 you state that you minimum event length is 20s whereas here you state it is 12 s.
These are two different post-processing steps. In the first step, we applied a duration threshold to the events obtained from the HMM classification, as suggested by Heck et al. (2018). In the second part, however, we applied a duration threshold to the median back-azimuth path obtained by MUSIC. This median path was calculated using a sliding window of 8 seconds. To obtain reliable results, it was thus necessary to use a second duration threshold value. We clarified this better in the text. (P.9 L.20)

*Heck et al. 2018a suggested that a detected event should have a minimum duration of 12 s to be considered as an avalanche. For the localization step, however, it was necessary to increase this duration because the window length used for the median smoothing filter was already 8 s long (Heck et al. 2018b).*
*To cover enough data points to use the minimal event duration as a reliable classification criterion, we therefore required a minimum length of 20 s for the back-azimuth path.*

p9, 6: What do you mean with "classes with 5 and 6 votes" what votes?
This term refers to the events that are detected by 5 sensors or 6 sensors of the array. We rephrased this sentence and removed the terms votes and classes, to avoid confusion. (P.11 L.15)

*A quarter of the events were detected by 5 sensors, a quarter by 6 and about half by 7.*

p9, 10: It that a good thing or a bad thing that you detect avalanches that are not listed in figure 4? E.g. does this mean that there are avalanches missing in figure 4 that should have been listed or are there completely different avalanches recorded in different areas and the only common thing is the huge amount of snow in that time period?

In principle, it is entirely possible that we detect avalanches with our monitoring system that were not recorded by visual observations in the area of Davos. However, at this point in the analysis, it remains unclear if the events automatically identified with the HMM correspond to

avalanches or not. We therefore manually went through all the 117 detected events to evaluate if the signal characteristics correspond to those typically seen for avalanches.

We changed the caption. (P.11)
*Avalanche released on 9 March 2017 at 06:47 used as training event for the classifier. a) time series for the 7 sensors. The red area indicates the part of the time series used as training event. b) corresponding spectrogram of the seismic time series.*

As already mentioned earlier, this refers the number of sensors the signal was detected at. We changed this term, since it is confusing here and throughout the text and figures.

We changed it.

This is due to the instrumentation used and the size of the avalanches that were monitored. We added some references in the introduction considering this small range of detection. (P.2 L.26)

*In the both studies of Lacroix et al. (2012) and Heck et al. (2018) less sensitive vertical component geophones were used for the seismic monitoring resulting in an avalanche detection of approximately 3 km.*

During the winter 2016 – 2017 we had slightly different meteorological conditions at the Dischma field site compared to the surroundings of Davos. In particular, there was less snow in the back of the Dischma valley resulting in fewer avalanches early in the season. Based on field observations and the images from the automatic cameras, we determined that in early February there were no medium-sized to large avalanches in the vicinity of our array.

The red area in Figure 5 is used as the training event, so we manually defined this area. The gray area in Figure 7, however, shows the part of the time series, which was identified as an event by the hidden Markov model. Therefore these two areas have a different meaning.

We agree with the reviewer that typically signals generated by airplanes either have clear overtones or at least a clear Doppler effect in the signal. However, based on our experience with seismic data from both arrays above Davos, we are confident that the signals shown in Figure 7 are generated by airplanes. We do not know yet why airplanes generate such signals, and we have not identified a clear pattern, which explains their presence. Perhaps it is related to the altitude or distance of the airplane, or perhaps specific atmospheric conditions are

responsible. Nevertheless, we have seen multiple signals like these recorded at both arrays and we are confident that these signals are generated by airplanes.

P13, figure 8: What is the unit of the normalised time and how is it calculated? Do the events have the same length or did you just stretch/ squeeze them to fit in between 0 and 1?

In Figure 8, time is normalized by the duration of each event. The normalized time therefore has no -units. 0 corresponds to the start of the event and 1 to the end of the event. The duration of the events can therefore not be determined from this plot as each event has a different duration. We chose this representation to better highlight similarities in the temporal behavior of the features for different events.

p13, 6: one ")" too much

We changed it.

p13, 8: one ")" too much

We changed it.

p13, 11: Do you know what these 37 other avalanche like events might be? Maybe these are just avalanches along an unexpected path or longer paths?

Based on a visual inspection of the seismic time series and spectrograms we assumed that we have several sources for these signals. For one part, some airplane signals were still present in the classification results after the combined array classification and signals produced during periods of strong winds.

p13, 16: It sounds to me a bit like you remove events until you end up with back azimuths or locations you would like to get.

We did not filter specific back-azimuth angles. We only require the derivative of the median back-azimuth path to fall within certain threshold values (i.e. the source of the signal should not vary too much within a certain period of time), independent of the values of the back-azimuth angle. The fact that the remaining events all had a mean back-azimuth pointing towards nearby avalanche slopes therefore confirms that this method is suited to identify signals likely generated by avalanches.

p14, figure 9: How do you know that these are airplanes?

During the last years we analyzed the continuous seismic data recorded during the last winter periods. These types of signals were often recorded almost simultaneously at both field sites at any time of the winter season, even in the beginning, when there was no snow at the field sites. We also compared the times of some signals with flight information and were able to identify the airplanes. See also comment above.

p15, discussion: I find the discussion a bit repetitive with respect to the rest of the manuscript. Many points seem to have been made already in the rest of the text. Also my impression is that they barely refer to work of others in the discussion i.e. papers that are not lead by "Heck" or "Hammer".

We agree with the reviewer that some parts of the discussion were redundant. We therefore partly rewrote the discussion to avoid this. However, since we are not aware of many studies which tackle the automatic detection of mass movements (avalanches or other) in continuous seismic data, there are not many other studies which are relevant. (P.20 L.5)

*Apart from HMMs, several other machine learning techniques are suited to classify signals in seismic data. It is possible to use a convolutional neural network for earthquake detection and location (Perol et al., 2018) or to pick the P-wave arrival of seismic wave fields (Ross et al., 2018). Comparable to the classical HMM approach, these studies rely on large pre-labelled training data sets. Another approach is the so-called Random Forest classifier, which can be used to discriminate seismic waves (Li et al., 2018). Automatic classification approaches are also suitable to differentiate between earthquakes and quarry blasts (Hammer et al., 2013) or to characterize larger rockfalls (Dammeier et al., 2016). Further mass movements, such as landslides, can also be identified in the seismic data based on automatic classification approaches (Esposito et al., 2006; Hibert et al., 2014; Provost et al., 2016). The automatic classification of avalanches yet remains a difficult task. Rubin et al. (2012) used several machine learning algorithms to identify avalanches in seismic data and compared the results obtained with the different approaches. With all methods a high probability of detection was achieved, but the number of false alarms was too high. A recent study by Heck et al. (2018a) showed that HMMs are a suitable tool to detect avalanches, but there is still a need for additional post-processing steps. In the work presented here we confirm that HMMs in combination with further post-processing steps provide reliable classification results.*

p15, figure 10: change to that the legend is not overlapping the bar any more
We changed the legend.

p16, figure 11: "for avalanche event" replace with "for an avalanche event"
We rephrased it.

Figure 11a: I don't understand to what part of the figure you refer to with "solid part". Beneath what threshold?
The solid part refers to the solid line in the polar plot.

Figure 11b: is this really the derivative of the angle (y axis label) or derivative of the back-azimuth path (caption)? To me this figure seems to show the "angle" or "back azimuth" during, before and after the avalanche event with very stable back azimuths during the event and larger scatter afterwards.
This plot shows the approximate derivative of the angle (due to the small amount of data points). It was calculated using the angdiff function of matlab and the step size dt between the data points angdiff/dt. We changed the label to clarify this. (P.17)

p17, figure 12: so there are 100 visually observed avalanches in Davos but you could detect only 20? Were you too far away or was this recorded but not classified as event? Move the legend so that it does not overlap with the bars
Indeed the differences in the number of avalanches relate to the size of the area that is monitored. The visual observations are made for an area of about 175 $km^2$, while with the seismic system only avalanches within a radius of 3 km can be monitored (~30 $km^2$). We changed the legend of the figure as suggested. (P.18)

p17, 1: "closer" replace with "closer to"?
We changed it.

p17, 8-10: First you say that you could confirm no avalanche visually, but in the next sentence you state that "another 12" events were identified. Were they identified in a different way i.e. not visually or is there an error in the sentence?
The events were not visually identified, but by inspecting the seismic time series and analyzing the events in more detail. Based on the results, these 12 events were identified as avalanches.
We clarified this point in the manuscript. (P.20 L.3)

*However, Heck et al. 2018B manually identified 13 avalanches during 9 and 10 March 2017, 12 of which were automatically identified with the approach presented here.*

P18, figure 13: How do you know to what distance the duration of the event corresponds to?
The duration and back-azimuth of the event are automatically determined with our method. We do not have any information on the distance of the source. Perhaps the reviewer misunderstood the results shown in Figure 13 and we therefore changed this figure. In the polar plot the direction of the lines indicates the direction of the back-azimuth, the thickness of the lines indicates the duration of the event. Thin lines correspond to short events, thick lines to long events. The thicker the line, the longer the event. The color code shows the release time of the avalanche. (P.19)

P18, 4: number of votes: in my opinion it would be better to replace "vote" with something like "detections on sensors" or similar.
As already mentioned above, we changed this term, since it is confusing.

p18, 12: the overall feature behavior from distance airplanes... "was" not "were"
We changed it.

p19, 9: remove "really". Based on the 5 events that were possible to locate, it is apparently possible to detect some avalanches on both arrays.
We changed it.

p19, 9: I am not sure I fully agree. It is not possible to record an avalanche at 14 km distance if it couples to the ground sufficiently or is large enough?
It might be possible to record avalanches at distances > 14 km. However, as previously mentioned, this is only for large and catastrophic avalanches which were not observed during our study period.

P19, 10: "since distance" replace with "since the distance"
We changed it.

p19, 10: I am not sure where installing two arrays at 2-3 km distance would help. They would then pick up the same avalanches, and hence "events recorded at both arrays" are then not a valid criteria any more to find falsely classified earthquakes or airplanes.
We agree. Two arrays will mainly improve the localization.

p19, 11: "improving" replace with "improve"
We changed it.

p19, 22-24: Can you not locate airplanes and earthquakes with the array because the frequency content is different? So if the MUSIC method is perfectly suitable of detecting

avalanches, why should one go through the hassle of finding a exemplary event, the need of having two arrays and then removing a lot of false detections? Rather than using the output from the array method to detect events?
We performed several test during this study considering the calculation time of the MUSIC method. Depending on the frequency range of the signals, the duration of the calculation varied. However, based on the actual code, it was not possible to apply the MUSIC method in near real-time. If we skipped the classification process at all, we would have to analyze all data remaining after the pre-processing step. It would take too much time to calculate the MUSIC results for all these events, even if we reduced the frequency range to a minimum. The classification process, however, has proven to be faster and allowing us to perform a near real-time classification.

p19, 26: typo in "theses"
We changes it.

p19, 30: typo "form"
We changed it.

p19, 32: "avalanches were released" instead of "avalanches released"?
We changed it.

p20, 5: Why is it that costly? Can the processing be sped up?
At the moment the algorithm is written in MATLAB. In this algorithm, the covariance matrix of small time windows is calculated. It might be possible to speed up the calculation process by using computers with more RAM, since matrix calculations depend on the size of RAM. Unfortunately, we are not computer engineers and our knowledge is too limited to answer the question.

p20, 14: "be still needed" replace with "still be needed"
We changed it.

p21, references. There are 11! referrals in the text to a not published paper (Heck et al. 2018b). Can the authors provide the manuscript in order to cross-check e.g. the content?
The paper is now accepted and accessible online: https://doi.org/10.1093/gji/ggy394

---

## Author Comment (AC2) · 17 Oct 2018

**Reply to reviewer 2**

We thank the reviewer for the constructive comments. Below we reply in detail.

1.) I'd like the author to better address the possible technical limitations of their methods, in particular the field deployment and the near-real-time application of the classification methods based on two stages.
We adapted the Discussions section and added a more detailed description of the limitations of our method (P. 22 L.5).

*Although we were able to identify one major avalanche activity period in the winter season 2016-2017, the method presented here has its limitations. Based on the sensors used for the automatic monitoring, we identified avalanches within a range of 2 – 3 km. However, by using more sensitive sensors, e.g. seismological broadband stations, the detection range of avalanches can be increased, even up to 30 km for very large avalanches (run-out distance > 2 km) (Hammer et al. 2017). However, it is difficult to deploy such sensors in mountains terrain, since these stations require existing infrastructure (e.g. electricity, storage room in a hut), which is typically not available at remote locations. In addition, the last post-processing step requires a second array. Hence low energy systems with less sensitivity proved to be the best solution. Furthermore, the limited power supply at the field sites also prevents performing first processing steps directly at the field sites and hence limits the possibility of near real-time analysis. However, it is possible to overcome this problem by designing special hardware for this particular task.*

2.) The network geometry has a strong impact on the success rate of this latter criterion, could you add some details on that?
Unfortunately, the resolution of our seismic array was limited due to the instrumentation used. Lacroix et. al 2012, e.g., used a seismic array with larger distances between the sensors resulting in a better resolution for seismic waves. They could use beam-forming methods to calculate the source direction. We assume that with a larger interstation distance the resolution can be improved. With a larger array, a new comparison between beam-forming and MUSIC would be required.

3.) I have the impression that this second stage can be surely useful to recognize earthquakes but it probably needs a calibration for anthropic sources.
It depends on the type of anthropogenic source. Helicopters, for example, have a very characteristic spectrogram and Doppler effects can be observed at all times. For airplanes, this is a little bit more complicated. Due to the different types of airplanes (turbine or propeller) and the different flight altitudes, signals vary. But most of the time, a dominant frequency is visible as well as Doppler effects. However, the airplanes, which were considered as avalanches by our algorithm, did not have a dominant frequency. We expect that these airplanes were flying at a larger distance to the seismic array and due to the topography signals were naturally filtered.
We gained this knowledge over the past years in a long learning step. We agree that calibrations for anthropogenic sources may provide more reliable classification results.

4.) In addition, technical limitations in such extreme environments like high Alpine areas (e.g., data transmission) can be a possible trivial but concrete limitation for a real time application.

That is correct. The biggest problem we have is the computational power at the field sites. At the moment, we rely on Raspberry Pis which are mainly used for data storage and data transfer purposes. Data are transmitted and then mainly processed at our institute. A first improvement would be, to establish a fast and stable wireless link to the arrays to provide good data transfer. Second, better hardware with a low energy consumption would also be a great advantage. However, hardware capable to perform our calculations near real-time cannot run solely on solar power and batteries.

5.) The application of the proposed methodology on another dataset gathered on another test site would be of great interest for the reader. For instance, is it possible to run the methods the other way round, testing it on the other array currently used for the second classification step? Of course it would be possible to perform the classification task the other way round. However, due to some technical issues, we only recorded with two sensors at the second field site. As we showed in a previous study (Heck et. al 2018a), best classification results with the HMMs are obtained by only considering events classified as an avalanche by at least five sensors. We could perform the classification with only two sensors, however, we expect too many false alarms. Furthermore, it is impossible to determine the source direction of the signals based on only two vertical geophones. Hence, the localization step could not be used to confirm or neglect detected events as avalanches.

6.) Visual observations are used as validation, could the authors add some information about that? Which are the observation sources? How is compiled the avalanche catalog? If available, an image of one reference event could be also useful to show the test site. We have installed several automatic cameras at the field sites, still, we could only determine the exact release time for two avalanches. For the remaining events, especially for the main avalanche activity period in March, we narrowed the release times down to a 24-hour window. This was due to the fact, that most avalanches released during snow storms or at night when the visibility was bad.
In addition to the cameras, we relied on the avalanche data base compiled by the avalanche warning service in Davos. They monitor the avalanche activity for the region of Davos (~ 175 km²) and also use information from voluntary observers.
We included a picture shortly after the avalanche period in March taken during a field survey.

7.) Figure 2, it would be useful to add a map with terrain information (slope, morphology, etc).
Since the interstation distance between the sensors is two small, it is nearly impossible to show additional terrain information in these particular figures. Hence we added two additional figures, each showing additional information of the field sites also including the location of all instrumentation, at a lower scale (P.6).

---

## Referee Report (RR1)

Review of
**Automatic detection of avalanches using a combined array classification and localization**
by Heck et al.

1st Revision

The two reviewers of the original submission pointed out several specific issues within the manuscript. The authors responded to all of these comments, yet sometimes it is hard for me to check if or how those replies made it into the revised version of the manuscript. Additionally I think the manuscript still needs work before it can be accepted for publication. Please find my detailed comments below.

Best regards,
Florian Fuchs
* * *
Handling of reviewer comments:

The authors reply properly to all reviewer comments. However, especially when answering to Reviewer #1 comments on pages 12 and later it's not clear anymore if those replies were integrated into the manuscript. I do support all of the reviewers comments and questions and I do see that the authors know how to respond to those. But I strongly suggest to implement all of them – at least briefly – into the manuscript. The same holds for almost all comments by Reviewer #2.

Please insert all of those replies to the text (at least briefly) and indicate all the changes in the rebuttal letter. Otherwise, it's hard to follow, not having done the 1st round of reviews.
* * *
Additional comments:

Although all/many comments from the first round of reviews were already taken care of, I must unfortunately admit that I still had a hard time reading the manuscript. I do believe that work itself is interesting and the findings are worth reporting. Yet, the manuscript is not easily comprehensible in the current shape. Mainly, I am missing a clear and concise structure and more precision in the wording and figures. I also suggest to make use of the Copernicus English grammar and spellchecking service.

General structure and ease of reading:
- Please be more precise throughout the entire manuscript. When you say "high", "low", "good", "poor", "better", "most", "large" please try to give values, if possible. E.g., what number of percentage can be considered a "good" classification result? When you speak of "features" that "change" and are "common" please describe specifically which features you mean and how they change.

- Your chain of processing kind of gets obscured throughout the manuscript. I'd suggest that somewhere you briefly list your work flow. Figure 5 somewhat tries to summarize this, but I think text would help here. Additionally, Figure 5 could use some instructive labels, e.g. you

could indicated the length of the data windows. The panels "pre-processing" and especially "post-processing" could indicate what's actually done. E.g. that post-processing is the MUSIC beamforming.

- I am missing a short subsection on "post-processing" in section 3, "methods". You repeatedly emphasize the need for "post-processing" but it's not clear what this is.

- In principle it is a good idea to have dedicated sessions on methods and results (sections 3 and 4 in this manuscript). Yet, you mix methods, observations, interpretation and references repeatedly. In the "Methods" section you should be as brief and precise and necessary. You should not evaluate the results of other work here, but only briefly repeat the main points you make use of. All the rest is better placed in the discussion section. Likewise, in the results section (4) you repeatedly evaluate the quality of the results (this should be done in the discussion section) or introduce new steps in the processing. Please double-check to clarify.

- It's difficult to track how many events you remove during the different processing steps and how many events actually remain as final detections. Maybe a table listing the number of events and how many get discarded by each processing step would help.

Discussion section:
- I agree with Reviewer #1 that the "Discussion" section in the current state is rather a repetition and summary of the previous chapters. This needs to be changed. Here I'd like to see you discuss the benefits and limitations of your methods. E.g. it is very interesting to read that the sensor installation itself already has a huge impact on the classification results. Why? What else can influence the classification that strongly? The airplane signal could also be discussed here (or in the supplemental, see below), as reviewer #2 points out the strong difference to other observed airplane signals. What about anthropogenic signals? Are there roads/cars nearby?

- Most importantly, the choice of the training event should be discussed, as it surely has a huge impact. For example why did you only choose a part of the avalanche signal in Figure 7 as the training event? Half of the signal seems to be missing … You may not have the time and patience now to carefully double-check the performance of your routine based on different training events, but this would of course be desirable. Do you maybe at least have some experience from other datasets that you can report on? Why can't you simply use more than one training event?

- Could you think of other "features" that could help to distinguish avalanches from airplanes and earthquakes? After all, the ones you use don't seem to do the job. I'd personally like to see you speculating here …

- Obviously, broadband sensors will not necessarily improve your data quality, neither will they automatically detect more distant avalanches. This needs to be rephrased. Only in the rare case of huge, catastrophic events – which generate long period seismic radiation, in contrast to the small local ones – they might be an advantage over the short-period geophones. The fact that "common" avalanches can only be detected within few km distance is probably due to the weak seismic signal they generate, and the only chance to improve the data quality is to have more sensors (signal-to-noise ratio) closer to the events (less attenuation). Of course, this is not always possible.

Efficiency of computations:

- When discussing the "speed of processing" you refer to a "standard 8 core processor with 16GB RAM". It may seem picky now, but do you actually make use of all the 8 cores? Is there some kind of parallelization involved in your processing? If yes, please comment on this, if not I think the community usually refers to "a standard personal desktop/laptop computer" to indicate that no supercomputing powers or high-level workstations are required. Similarly, do you actually need the 16GB RAM? If yes, what for?

    Reviewer #2 also pointed this out and it's actually an interesting point. In fact, probably the computing power wouldn't really matter and you would not have to comment on it, if data were only processed "off-field" in some data center. However, In your reply, you indicate that some of the processing is done on-site in the field – this of course strongly limits computational power and is a very interesting and crucial point that is not mentioned at all in the manuscript. Please include this in the Instrumentation/Methods section! This will also clarify why you perform some of the processing steps and why computation time is crucial.

Figures:

- There are a lot of Figures, which complicates the reading. I suggest to e.g. somehow merge Figures 6, 8, 12 and 14 as they all show the same information. If all the panels were shown below each other, a comparison of the observations would be easier.

- Similarly, maybe Figures 2 and 3 could be merged.

- Please highlight the avalanches in Figure 4.

- Figures 9 + 11 are not relevant for the understanding of the text and I suggest to move those to the supplemental material. Reviewer #2 raised doubts about the origin of the airplane signals, since they look different in other studies. The authors claim to be certain about their interpretation. This point might also be discussed in the supplemental material, as it's not crucial for the understanding of the main text.

---

## Author Response (AR2)

Reviewer 2:

The author describes an approach for a detection method for avalanches based on hidden Markov models, applied on a seismic array. I' am wondering about the high computational and installation effort (two arrays) needed for this detection method, which does not make sense for practical applications. Also the false alarm ratio is relatively high, although a complicated detection method is used. However, the method and the results are well presented and the work has a scientific relevance for seismic signal processing , so I recommend to accept this article with minor revision:

Page 1, line 11: "small changes of source direction" - this depends on the location array - avalanche path
This is correct. However, since our array is located at the valley bottom, we expect only small changes for the back-azimuth. We clarified it in the abstract.
*Since snow avalanches recorded at our arrays typically generate signals with small changes in source direction, events with large changes were dismissed as false detections.*

Page 2, line 13-18: Might mention that infrasound detection methods for avalanches currently shows better results than seismic detection. Of course this depends on the avalanche type, so powder avalanches produce higher infrasound amplitudes, while wet snow avalanches generate higher seismic signals. You can also notice, that a combination of both technologies might result in a better detection performance.
We added a few sentences in the introduction to address this issue.
*A recent comprehensive study on the performance of these systems has shown that in the absence of major topographic barriers, infrasound avalanche detection systems relying on array processing techniques are well suited to reliably monitor larger avalanches up to a distance of 3 to 4 kilometers (Mayer et al. 2018).*

Page 3, line 10: "natural frequency" instead of "eigenfrequency"
We changed the term.

Page 4, sec. 3.1: What does this threshold mean for the possible detectable avalanche size? Have you thought about using the common STL/LTA method?
This threshold was only used to reduce the amount of data to process. It is similar to the STA/LTA method, however, since we are not interested in the exact onset of seismic signals (which is a big advantage of the STA/LTA method, especially for earthquake detection), a much simpler amplitude threshold was sufficient.

Page 7, line 25: I'm wondering, that you can find one representative avalanche event for training, which can be used for the whole winter season. Normally there a large differences at the signal pattern for different avalanche types (powder to wet snow avalanches).
In previous studies, we investigated using different avalanche signals to represent different classes (i.e. dry and wet-snow avalanches). However, this approach did not improve the classification results. Signals from different types of avalanches have some distinct characteristics: wet-snow avalanches generally generate longer signals (these avalanches flow more slowly) and higher amplitudes (larger mass often flowing on the bare ground). Nevertheless, when using HMMs for the classification, the duration of the signal and the maximum amplitude are not relevant  and there is no need to implement specific avalanche classes.

Page 8, line 8: Is not it theoretical possible that an avalanche occur right between this two arrays and is than registered by both?

The maximum distance for an avalanche to be detected is around 3 km. Since both arrays are seperated by about 14km, avalanches occurring right between those arrays are likely not detected at all. Furthermore, it is possible that avalanches releases simultaneously at both array. However, we assume this probability to be rather low.

Page 9, line 20-24: What does this minimum event duration mean for detectable avalanche sizes?

It is clear that imposing a duration threshold for the detections does not allow us to investigate small avalanches. However, due to a lack of ground truth data, we did not investigated the influence of avalanches size. Previous work has shown that signal duration relates to avalanche size. However, as powder avalanches travel at higher velocities than wet snow avalanches, a long duration of the signal might indicate a large and fast powder avalanche with a long runout, or a slow wet snow avalanche with a shorter runout and size. Thus, we cannot comment specifically on what avalanche size is excluded due to our minimum duration threshold.

Sec. 5: A graphic comparing the number of detected avalanches, false alarms (maybe separated for every detection criterion) and also the number of missed events for the whole season would be useful. Especially I missed a detailed description about the missed events.

We included an additional Table containing these numbers.

Page 20, line 10: Might you can also note literature about seismic detection of debris flow/debris flood - this are sometimes similar to the detection methods for avalanches.

We have added more references in the Discussion section related to other types of gravitational mass movements.

Page 21, line 12: Efficient for your situation, but the need of two different arrays is not a "efficient approach".

We now address this point in the Discussion section:

*Our suggested workflow requires two arrays to eliminate falsely classified events by finding co-detections. This is clearly a limiting factor as it increases the cost for the instrumentation as well as deployment and maintenance time.*

Review of
**Automatic detection of avalanches using a combined array classification and localization**
by Heck et al.

1st Revision

The two reviewers of the original submission pointed out several specific issues within the manuscript. The authors responded to all of these comments, yet sometimes it is hard for me to check if or how those replies made it into the revised version of the manuscript. Additionally I think the manuscript still needs work before it can be accepted for publication. Please find my detailed comments below.

Best regards,
Florian Fuchs

Handling of reviewer comments:

The authors reply properly to all reviewer comments. However, especially when answering to Reviewer #1 comments on pages 12 and later it's not clear anymore if those replies were integrated into the manuscript. I do support all of the reviewers comments and questions and I do see that the authors know how to respond to those. But I strongly suggest to implement all of them – at least briefly – into the manuscript. The same holds for almost all comments by Reviewer #2.
Please insert all of those replies to the text (at least briefly) and indicate all the changes in the rebuttal letter. Otherwise, it's hard to follow, not having done the 1 st round of reviews.

Additional comments:
Although all/many comments from the first round of reviews were already taken care of, I must unfortunately admit that I still had a hard time reading the manuscript. I do believe that work itself is interesting and the findings are worth reporting. Yet, the manuscript is not easily comprehensible in the current shape. Mainly, I am missing a clear and concise structure and more precision in the wording and figures. I also suggest to make use of the Copernicus English grammar and spellchecking service.

We would like to thank the reviewer for the helpful comments. Based on these comments, we substantially changed the manuscript to improve the structure and clarity of the text. We have also (briefly) included many of our replies of the first round of reviews in the discussion.

General structure and ease of reading:
- Please be more precise throughout the entire manuscript. When you say "high", "low", "good", "poor", "better", "most", "large" please try to give values, if possible. E.g., what number of percentage can be considered a "good" classification result? When you speak of "features" that "change" and are "common" please describe specifically which features you mean and how they change.

We checked the manuscript and changed the wording in places where it was ambiguous.

- Your chain of processing kind of gets obscured throughout the manuscript. I'd suggest that somewhere you briefly list your work flow. Figure 5 somewhat tries to summarize this, but I think text would help here. Additionally, Figure 5 could use some instructive labels, e.g. you could indicated the length of the data windows. The panels "pre-processing" and especially "post-processing" could indicate what's actually done. E.g. that post-processing is the MUSIC beamforming.

We completely restructured the methods and results sections to more clearly convey how the suggested signal processing workflow works.

- I am missing a short subsection on "post-processing" in section 3, "methods". You repeatedly emphasize the need for "post-processing" but it's not clear what this is.

We added a section to more clearly explain the different stages in the post-processing.

- In principle it is a good idea to have dedicated sessions on methods and results (sections 3 and 4 in this manuscript). Yet, you mix methods, observations, interpretation and references repeatedly. In the "Methods" section you should be as brief and precise and necessary. You should not evaluate the results of other work here, but only briefly repeat the main points you make use of. All the rest is better placed in the discussion section. Likewise, in the results section (4) you repeatedly evaluate the quality of the results (this should be done in the discussion section) or introduce new steps in the processing. Please double-check to clarify.

As mentioned before, we completely restructured the methods and results sections. Furthermore, we moved some sections of the text to either the introduction or the discussion.

- It's difficult to track how many events you remove during the different processing steps and how many events actually remain as final detections. Maybe a table listing the number of events and how many get discarded by each processing step would help.

We added an additional table showing the number of detection and false classification for each processing step.

Discussion section:
- I agree with Reviewer #1 that the "Discussion" section in the current state is rather a repetition and summary of the previous chapters. This needs to be changed. Here I'd like to see you discuss the benefits and limitations of your methods. E.g. it is very interesting to read that the sensor installation itself already has a huge impact on the classification results. Why? What else can influence the classification that strongly? The airplane signal could also be discussed here (or in the supplemental, see below), as reviewer #2 points out the strong difference to other observed airplane signals. What about anthropogenic signals? Are there roads/cars nearby?

We rewrote most of the Discussions section and now also explicitly address the airplane signals that were falsely classified as avalanches.

- Most importantly, the choice of the training event should be discussed, as it surely has a huge impact. For example why did you only choose a part of the avalanche signal in Figure 7 as the training event? Half of the signal seems to be missing ... You may not have the time and patience now to carefully double-check the performance of your routine based on different training events, but this would of course be desirable. Do you maybe at least have some experience from other datasets that you can report on? Why can't you simply use more than one training event?

As mentioned in the discussion section now, it was best to neglect the coda of the avalanche signal and only use the part of the signal where energy increases up to the first maximum of the signal. We also investigated using different sections of the avalanche signal without improving the classification results. In the past, we have investigated using different training events for dry- and wet-snow avalanches to improve our classification results. However, such an approach did not improve the results at all and typically resulted in more falsely classified events. While we did not conduct a comprehensive investigation on the influence of the training event, our ad-hoc testing has shown that the influence is rather limited. Furthermore, we also wanted to highlight that it is possible to use this classification approach with only one training event.

- Could you think of other "features" that could help to distinguish avalanches from airplanes and earthquakes? After all, the ones you use don't seem to do the job. I'd personally like to see you speculating here …

We investigated using other and more features in our recent work (Heck et al. (2018)). In the end, the feature combination used for the classification in the current work was the best suitable for the classification task.

- Obviously, broadband sensors will not necessarily improve your data quality, neither will they automatically detect more distant avalanches. This needs to be rephrased. Only in the rare case of huge, catastrophic events – which generate long period seismic radiation, in contrast to the small local ones – they might be an advantage over the short-period geophones. The fact that "common" avalanches can only be detected within few km distance is probably due to the weak seismic signal they generate, and the only chance to improve the data quality is to have more sensors (signal-to-noise ratio) closer to the events (less attenuation). Of course, this is not always possible.

We agree with the reviewer that changing the instrumentation is not likely going to improve the data quality nor the detection range. We therefore removed this section.

Efficiency of computations:
- When discussing the "speed of processing" you refer to a "standard 8 core processor with 16GB RAM". It may seem picky now, but do you actually make use of all the 8 cores? Is there some kind of parallelization involved in your processing? If yes, please comment on this, if not I think the community usually refers to "a standard personal desktop/laptop computer" to indicate that no supercomputing powers or high-level workstations are required. Similarly, do you actually need the 16GB RAM? If yes, what for? Reviewer #2 also pointed this out and it's actually an interesting point. In fact, probably the computing power wouldn't really matter and you would not have to comment on it, if data were only processed "off-field" in some data center. However, In your reply, you indicate that some of the processing is done on-site in the field – this of course strongly limits computational power and is a very interesting and crucial point that is not mentioned at all in the manuscript. Please include this in the Instrumentation/Methods section! This will also clarify why you perform some of the processing steps and why computation time is crucial.

We did not perform an in-depth analysis of our computing time, nor did we try to optimize our algorithms for this. We only wanted to comment on this to show that the method could be used in near real-time and that the most costly analysis is the MUSIC method. We do not perform any processing on-site in the field, this must be a misunderstanding or some unclear comments in our earlier replies. Note that it is useful to have a multicore processor, since several tasks can be performed simultaneously (e.g. computing the features of all seven sensors at the same time). We now only very briefly comment on this at the end of the Discussion section.

Figures:
- There are a lot of Figures, which complicates the reading. I suggest to e.g. somehow merge Figures 6, 8, 12 and 14 as they all show the same information. If all the panels were shown below each other, a comparison of the observations would be easier.
- Similarly, maybe Figures 2 and 3 could be merged.

We merged and improved some of the figures

- Please highlight the avalanches in Figure 4.

We highlighted avalanches in this figure and also included the location of the cameras and the seismic array.

- Figures 9 + 11 are not relevant for the understanding of the text and I suggest to move those to the supplemental material. Reviewer #2 raised doubts about the origin of the airplane signals, since they look different in other studies. The authors claim to be certain about their interpretation. This point might also be discussed in the supplemental material, as it's not crucial for the understanding of the main text.

We do not agree with the reviewer that these figures are not relevant. In our opinion these figures clearly show the two main types of signals that were falsely classified. Furthermore, one can clearly see that the signals recorded at both arrays are very similar. This is particularly important for our airplane signals, which do not contain any signs of Doppler effect or clear overtones and are therefore rather unusual. We therefore kept these figures in the main text, but we merged them into one figure.

[revised manuscript text omitted]